# The actin modulator hMENA regulates GAS6-AXL axis and pro-tumor cancer/stromal cell cooperation

Roberta Melchionna[1], Sheila Spada[1,†], Francesca Di Modugno[1], Daniel D'Andrea[2,††], Anna Di Carlo[1], Mariangela Panetta[1], Anna Maria Mileo[1], Isabella Sperduti[3], Barbara Antoniani[4], Enzo Gallo[4], Rita T Lawlor[5], Lorenzo Piemonti[6], Paolo Visca[4], Michele Milella[7], Gian Luca Grazi[8], Francesco Facciolo[9], Emily Chen[10], Aldo Scarpa[5] & Paola Nisticò[1,*] (iD)

## Abstract

The dynamic interplay between cancer cells and cancer-associated fibroblasts (CAFs) is regulated by multiple signaling pathways, which can lead to cancer progression and therapy resistance. We have previously demonstrated that hMENA, a member of the actin regulatory protein of Ena/VASP family, and its tissue-specific isoforms influence a number of intracellular signaling pathways related to cancer progression. Here, we report a novel function of hMENA/hMENAΔv6 isoforms in tumor-promoting CAFs and in the modulation of pro-tumoral cancer cell/CAF crosstalk via GAS6/AXL axis regulation. LC-MS/MS proteomic analysis reveals that CAFs that overexpress hMENAΔv6 secrete the AXL ligand GAS6, favoring the invasiveness of AXL-expressing pancreatic ductal adenocarcinoma (PDAC) and non-small cell lung cancer (NSCLC) cells. Reciprocally, hMENA/hMENAΔv6 regulates AXL expression in tumor cells, thus sustaining GAS6-AXL axis, reported as crucial in EMT, immune evasion, and drug resistance. Clinically, we found that a high hMENA/GAS6/AXL gene expression signature is associated with a poor prognosis in PDAC and NSCLC. We propose that hMENA contributes to cancer progression through paracrine tumor–stroma crosstalk, with far-reaching prognostic and therapeutic implications for NSCLC and PDAC.

**Keywords** actin cytoskeleton; AXL; cancer-associated fibroblasts; GAS6; lung cancer
**Subject Categories** Cancer; Cell Adhesion, Polarity & Cytoskeleton

## Introduction

The tumor microenvironment (TME) is increasingly recognized as a source of novel therapeutic targets (Binnewies *et al*, 2018), and the identification of paracrine communication between tumor cells and cancer-associated fibroblasts (CAFs) is of great clinical relevance (Carr & Fernandez-Zapico, 2016; Gascard & Tlsty, 2016).

This general concept is of significance in non-small cell lung cancer (NSCLC), that despite the therapeutic efficacy of immune checkpoint blockade (ICB; Rizvi *et al*, 2015) still remains a tumor in which the stroma may hamper treatment efficacy. This scenario is even more detrimental in pancreatic ductal adenocarcinoma (PDAC; Kraman *et al*, 2010; Provenzano *et al*, 2012; Shi *et al*, 2019), a tumor which still lacks effective therapeutic options although estimated to become the second leading cause of cancer-related deaths by 2030 (Hoos *et al*, 2013; Rahib *et al*, 2014). CAFs are the main components of tumor stroma and exert tumor-promoting activities by modulating extracellular matrix (ECM), interacting with cancer cells (Olumi *et al*, 1999; Allinen *et al*, 2004; Toullec *et al*, 2010; Jacob *et al*, 2012; Gascard & Tlsty, 2016; Hammer *et al*, 2017), and through regulation of inflammation and anti-tumor immunity (Costa *et al*, 2018; Lakins *et al*, 2018; Elyada *et al*, 2019).

The recent identification of different CAF subtypes (Costa *et al*, 2018; Cremasco *et al*, 2018; Su *et al*, 2018) calls for the identification of the main players able to convert normal fibroblasts into pro-tumor CAFs and of the molecules used by CAFs to communicate with tumor cells promoting tumor growth and invasiveness.

Actin cytoskeleton dynamics and organization regulate cell–ECM and cell–cell contacts and have been also correlated with genome activity (Olson & Nordheim, 2010). We have recently demonstrated

1  Tumor Immunology and Immunotherapy Unit, IRCCS Regina Elena National Cancer Institute, Rome, Italy
2  Department of Medicine, Centre for Cell Signaling and Inflammation, Imperial College London, London, UK
3  Biostatistics and Scientific Direction, IRCCS Regina Elena National Cancer Institute, Rome, Italy
4  Pathology Unit, IRCCS Regina Elena National Cancer Institute, Rome, Italy
5  ARC-NET Research Centre, Department of Diagnostics and Public Health, Section of Pathology, University of Verona, Verona, Italy
6  Diabetes Research Institute, IRCCS San Raffaele Scientific Institute, Milan, Italy
7  Department of Medical Oncology 1, IRCCS Regina Elena National Cancer Institute, Rome, Italy
8  Hepato-pancreato-biliary Surgery Unit, IRCCS Regina Elena National Cancer Institute, Rome, Italy
9  Thoracic-Surgery Unit, IRCCS Regina Elena National Cancer Institute, Rome, Italy
10  Thermo Fisher Precision Medicine Science Center, Cambridge, MA, USA
  *Corresponding author. Tel: +39 0652662539; Fax: +39 0652662600; E-mail: paola.nistico@ifo.gov.it
  †Present address: Department of Radiation Oncology, Weill Cornell Medicine, New York, NY, USA
  ††Present address: MRC Centre for Neuropsychiatric Genetics and Genomics, Cardiff University, Cardiff, UK

that the actin regulatory protein hMENA controls the expression level of β1 integrin by affecting G-ACTIN/F-ACTIN, critical for the nuclear localization of the SRF co-factor myocardin-related transcription factor A (Di Modugno *et al*, 2018b).

hMENA (ENAH) belongs to the Ena/VASP family of actin regulatory proteins, which modulate cell–cell adhesion and cell migration (Bear & Gertler, 2009). The *ENAH* gene undergoes a splicing process generating multiple tissue-specific isoforms (Di Modugno *et al*, 2012). We have identified two alternatively expressed isoforms, the epithelial-specific/anti-apoptotic hMENA[11a] (Di Modugno *et al*, 2012; Trono *et al*, 2015) and the mesenchymal-specific/pro-invasive hMENAΔv6 (Di Modugno *et al*, 2012). hMENA/hMENAΔv6 regulate tumor growth factor TGFβ signaling and are crucial in TGFβ-mediated epithelial mesenchymal transition (EMT) (Melchionna *et al*, 2016). Clearly, hMENA and its isoforms play a central role in supporting malignant transformation and progression as demonstrated in different tumors (Di Modugno *et al*, 2006, 2018a,b; Gertler & Condeelis, 2011; Bria *et al*, 2014; Melchionna *et al*, 2016; Wang *et al*, 2017). We have proposed that hMENA isoform expression pattern is a powerful prognostic factor in NSCLC and pancreatic cancer, with a high overall hMENA (including hMENAΔv6) and low hMENA[11a] expression identifying patients with poor prognosis (Bria *et al*, 2014; Melchionna *et al*, 2016).

Here we asked whether hMENA may exert its role in cancer progression also by regulating CAF activation and their bi-directional communication with tumor cells.

We demonstrate that hMENA/hMENAΔv6 expression play a crucial role in the activation of CAFs derived from both NSCLC and PDAC patients and in their reciprocal interaction with cancer cells. Mechanistically, we identified that this hMENA-mediated pro-tumor function is attributable to its ability to regulate growth arrest-specific 6 (GAS6) in CAFs and AXL in tumor cells, sustaining the pro-tumoral paracrine GAS6-AXL axis, described as crucial in EMT, drug resistance, and immune evasion (Gjerdrum *et al*, 2010; Jokela *et al*, 2018; Ludwig *et al*, 2018).

PDAC and NSCLC patients show a worse prognosis when expressing high AXL-GAS6-ENAH gene expression compared with the combined expression of AXL and GAS6 and indicate the relevance of hMENA as both a prognostic marker and a potential therapeutic target.

## Results

### hMENA/hMENAΔv6 define a pro-tumor CAF activation state

Starting from the observation that stromal compartment of PDAC and NSCLC primary tumors showed in a number of cases a strong immunoreactivity for the Pan-hMENA antibody, compatible with CAF morphology, we evaluated whether hMENA and its isoform expression exert a role not only in tumor cells, but also in pro-tumor CAF biology.

We isolated CAFs from resected primary PDACs (P-CAFs) and NSCLCs (L-CAFs) (patient characteristics are shown in Appendix Table S1 and S2, respectively). The isolated CAFs exhibited typical feature of spindle-like mesenchymal cells and lacked the mutations found within primary tumors as revealed by NGS analysis for a panel of 22 genes in all CAFs used for functional studies

(Appendix Table S1 and S2). Furthermore, CAF IF analysis and qRT–PCR confirmed that these cells express CAF markers (i.e., FAP, PDGFRB) and are negative for EPCAM (Appendix Fig S1A–C).

Characterization of hMENA isoforms clearly showed that CAFs, as expected, were negative for the epithelial hMENA[11a] isoform (Fig 1A and Appendix Fig S2B and C) and for pan cytokeratin and E-CADHERIN (Fig 1), which are expressed in cancer cells (Ep-PDAC), and were immunostained by α-SMA and Pan-hMENA mAbs (the representative case PDAC#36 in Fig 1A).

The tissue specificity of hMENA splicing was confirmed by RT–PCR and WB analysis (Appendix Fig S2B and C) showing that CAFs expressed the hMENA (88 KDa), along with the mesenchymal-specific hMENAΔv6 isoform (80 KDa), but not the hMENA[11a] (90 KDa).

We then compared the expression level of hMENA/hMENAΔv6 between fibroblasts isolated from normal pancreatic tissues derived from transplant donors (P-NFs) and P-CAFs. WB analysis showed that both isoforms were expressed in P-CAF at higher level than in P-NF in the majority of cases evaluated (Fig 1B). Similar results were evidenced for L-CAF compared to lung normal fibroblasts (L-NF) (Fig 1C) and to paired "distal" fibroblasts (L-DFs) derived from non-"tumoral" tissue isolated at least 5 cm away from the tumor core (Appendix Fig S3A). Thus, hMENAΔv6 is expressed, although at heterogeneous level, in all our non-immortalized CAF cultures tested (Fig 1B and C and in Appendix Fig S3B), with the exception of #97 which was derived from a PDAC peritoneal metastasis (Fig 1B). Furthermore, we were able to isolate fibroblasts from a pancreatic serous cystadenoma #71 showing a very low hMENA/hMENAΔv6 expression (Fig 1B).

This heterogeneous hMENA expression in the stroma was also evidenced in the IHC analysis of primary NSCLC tissues (Appendix Fig S3C). To confirm that CAF present in primary tumor tissues overexpress hMENA/hMENAΔv6, we performed confocal analysis of NSCLC and PDAC tissues co-stained with Pan-hMENA and α-SMA antibodies, showing that Pan-hMENA decorates α-SMA-positive stromal cells and, as expected, also tumor cells which were α-SMA negative (Fig 2A and B).

Analysis of the Navab dataset (Navab *et al*, 2011) confirmed that in primary cultures of CAFs and matched non-malignant distal fibroblasts (NFs) from 15 resected NSCLC, *hMENA (ENAH)* expression correlated with the expression of *α-SMA (ACTA2)* and *FAP*, two of the main CAF markers (Appendix Fig S4).

Furthermore, in PDAC single-cell RNA-Seq data (Peng *et al*, 2019) we evidenced that *hMENA* expression levels are higher in fibroblasts and stellate cells, but not in immune cells, compared to the other stromal cell types (Benjamini–Hochberg adjusted $q$-values: Fibroblast, $q = 0.0002$; Stellate, $q = 1.5e-07$) (Fig 2C). Yet, *hMENA* expression levels are higher in Ductal 1 cells compared to the other tumor cells (Benjamini–Hochberg adjusted $q = 0.0013$). Similarly, single-cell RNA-Seq of the lung tumor microenvironment identified different stromal cell subtypes (Lambrechts *et al*, 2018). From this analysis we gathered results that *hMENA (ENAH)* is expressed (although heterogeneously among the clusters) at higher levels in fibroblasts compared to the other stromal cell types (Benjamini–Hochberg adjusted $q = 0.0022$) (Fig 2D).

To further detail the hMENA/hMENAΔv6 functional significance, we analyzed whether hMENA/hMENAΔv6 expression level was correlated to CAF activity by performing functional experiments in a

number of P-CAF and L-CAF and in P-NF and L-NF. We found that the higher expression of hMENA/hMENAΔv6 in CAFs with respect to P-NF and L-NFs correlated with a different ability to contract the collagen gel, a measure of their matrix remodeling capacity, and to

secrete and activate MMP-2 (Appendix Fig S5B–D). The activated phenotype of CAFs was confirmed by the increased expression of FAP in these cells compared to normal primary fibroblasts NFs (Appendix Fig S5A).

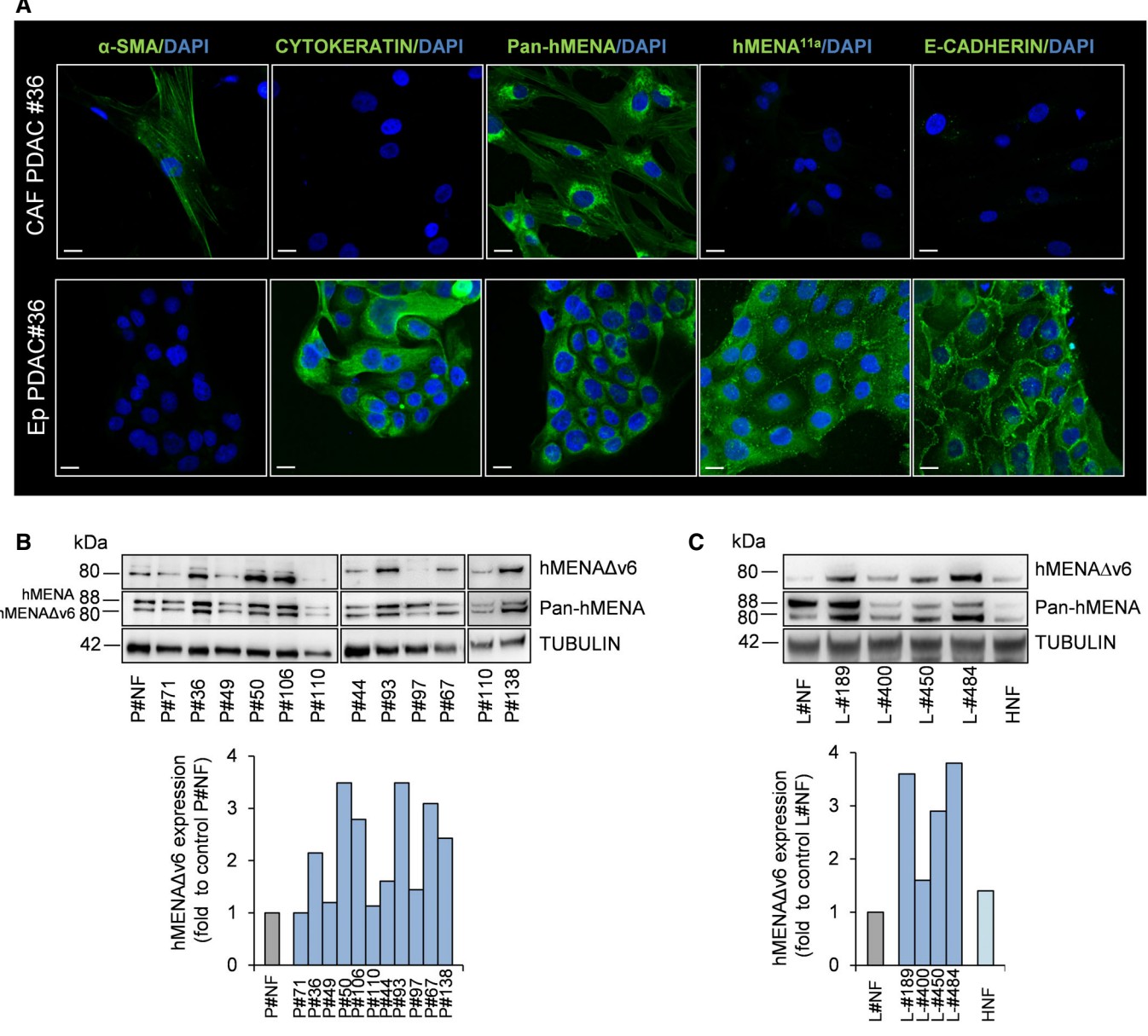

**Figure 1. hMENA isoform expression in CAFs.**

A   Representative images of immunofluorescence of α-smooth muscle actin (α-SMA), pan cytokeratin, Pan-hMENA, hMENA[11a], and E-cadherin expression in CAFs and autologous cancer cells (Ep-PDAC) obtained from enzymatically digested primary PDAC tissue of patient #36. Nuclei were stained with 4′,6-diamidino-2-phenylindole (DAPI). Scale bar: 20 μm.

B   Representative immunoblot (top) of hMENA/hMENAΔv6 expression levels (detected by Pan-hMENA mAb and by the specific anti-hMENAΔv6 antibody) in normal fibroblasts derived from transplant donor (P-NF), pancreatic serous cystadenoma #71, and cancer-associated fibroblasts (n = 10) obtained from primary PDAC tissues. Densitometry quantified data (bottom) of hMENAΔv6 expression. Quantification of P#110 is relative to the sample shown in the WB on the left.

C   Representative immunoblot (top) of hMENA/hMENAΔv6 expression levels (detected by Pan-hMENA mAb and by the specific anti-hMENAΔv6 antibody) in normal lung fibroblasts (L-NF), cancer-associated fibroblasts obtained from NSCLC tissues (n = 4), and normal dermal fibroblasts (HNF). Densitometry quantified data (bottom) of hMENAΔv6 expression.

Data information: Quantified data, are represented as fold change of hMENAΔv6/TUBULIN ratio with respect to control P-NF and L-NF (set as 1).
Source data are available online for this figure.

This hMENA-related CAF functionality (Fig 3) was evident when we silenced all hMENA isoforms by using a pool of three different siRNAs (sihMENA(t)), in CAFs with high hMENAΔv6 expression (P-CAF#36; P-CAF#138; L-CAF#189; L-CAF#484, Fig 1B and C). SihMENA(t) reduced the ability of CAFs to invade, as measured by Matrigel transwell invasion assay (Fig 3A) and to activate MMP-2 in

both L-CAFs and P-CAFs (Appendix Fig S6B). Moreover, we observed a significant decrease of the ability of CAFs to contract collagen gels in hMENA/hMENAΔv6 silenced CAFs compared to control CAFs (Appendix Fig S6A).

Of relevance when we overexpressed the hMENAΔv6 in P-NFs and L-NFs as well in CAF with low hMENAΔv6 expression

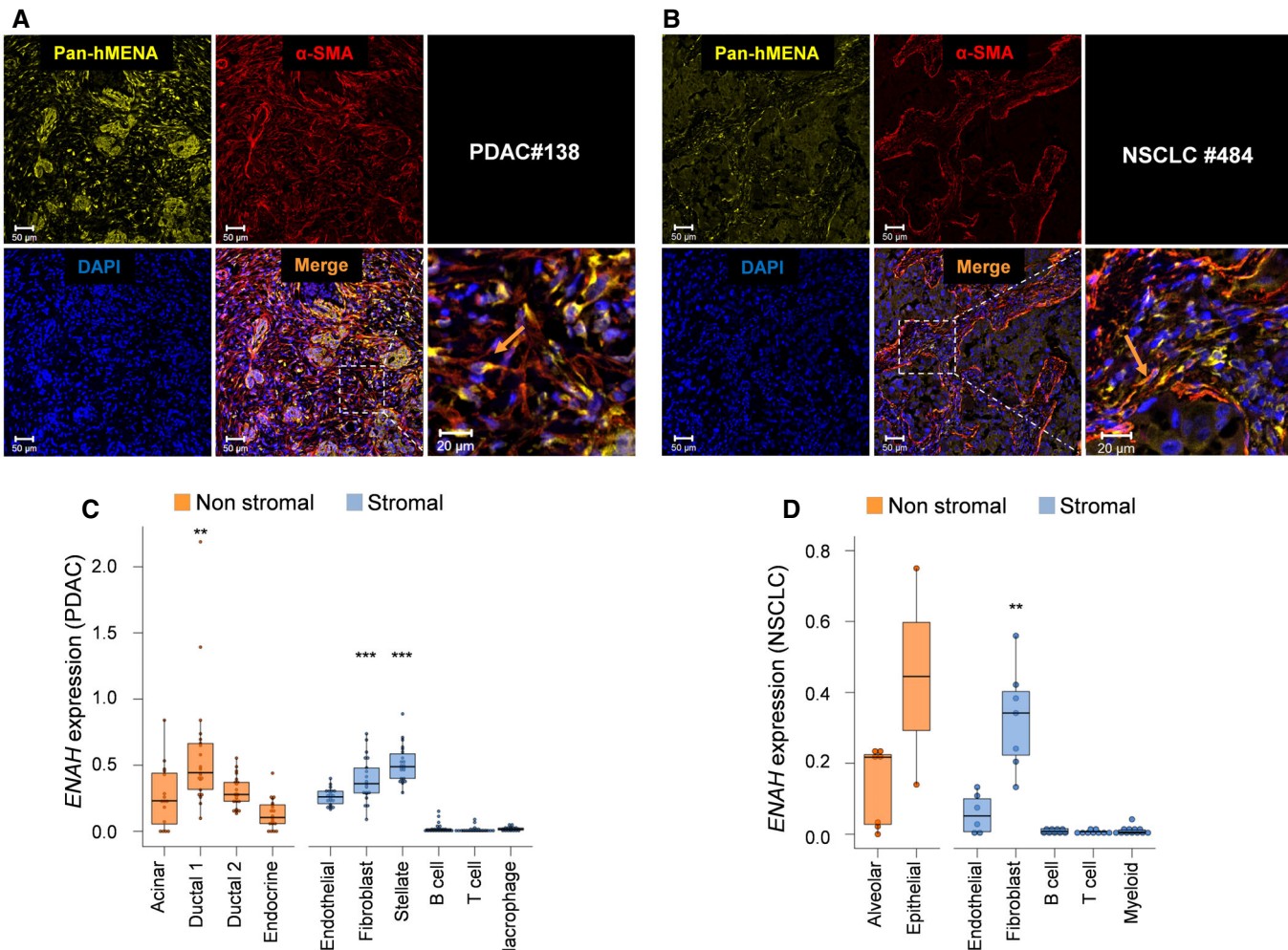

**Figure 2. hMENA expression in tumor and stroma of PDAC and NSCLC tissues.**

A   Representative images of immunofluorescence of Pan-hMENA (yellow) and α-SMA (red) in the primary PDAC tissue of patient #138 from whom high hMENAΔv6 CAFs were obtained. Nuclei were stained with 4′,6-diamidino-2-phenylindole (DAPI). Scale bar: 50 μm. The inset of the dashed area is provided on the right as a zoomed-in and cropped fluorescence image. Scale bar: 20 μm. αSMA-positive CAFs are also positive for Pan-hMENA (arrow).

B   Representative images of immunofluorescence of Pan-hMENA (yellow) and α-SMA (red) in NSCLC case #484 from whom high hMENAΔv6 CAFs were obtained. Nuclei were stained with 4′,6-diamidino-2-phenylindole (DAPI). Scale bar: 50 μm. The inset of the dashed area is provided on the right as a zoomed-in and cropped fluorescence image. Scale bar: 20 μm. As in A, α-SMA signal is evident in stromal cells which are also positive for Pan-hMENA (arrow).

C   Boxplots showing the mean mRNA expression of ENAH gene (hMENA) for 22 patients in the 10 groups of cell types (Peng et al, 2019; total number of cells: 38,487, mean of cells for patient: 1,603). Shown in each boxplot are the median value (horizontal line), 25th–75th percentiles (box outline), and highest and lowest values within 1.5× of the interquartile range (vertical line). Expressions from each cell-type group were compared to all other groups by using Mann–Whitney U-test (two-sided) and P values were adjusted for multiple testing using the Benjamini–Hochberg method. Stromal cell-type groups with significantly up-regulated ENAH expression respect to other stromal groups are: Fibroblast, ***q = 0.0002; Stellate, ***q = 1.5e-07. Non-stromal cell-type groups with significantly up-regulated ENAH expression with respect to other non-stromal groups are: Ductal 1, **q = 0.0013.

D   Boxplots showing the mRNA expression of ENAH gene (hMENA) in the seven groups of cell types from (Lambrechts et al, 2018) (total n = 52). Shown in each boxplot are the median value (horizontal line), 25th–75th percentiles (box outline), and highest and lowest values within 1.5× of the interquartile range (vertical line). Cell expression from each group were compared to all other stromal/not-stromal cells by using Mann–Whitney U-test (two-sided) and P values were adjusted for multiple testing using the Benjamini–Hochberg method (Fibroblast group vs other stromal groups, **q = 0.0022; Other comparisons, q > 0.05).

(P-CAF#110 and L-CAF#400, Fig 1B and C), we found that the non-tumoral fibroblasts and CAFs hMENAΔv6 low increased their functional activities (Fig 3B and Appendix Fig S6C).

Collectively, these data point for the first time to our knowledge to the role of hMENA/hMENAΔv6 as marker of pro-tumor CAF activation state.

## hMENA is crucial in the cooperativity between tumor cells and CAFs

It is well established that CAFs promote tumor progression and invasion in various cancers through the activation of paracrine signaling (Gascard & Tlsty, 2016).

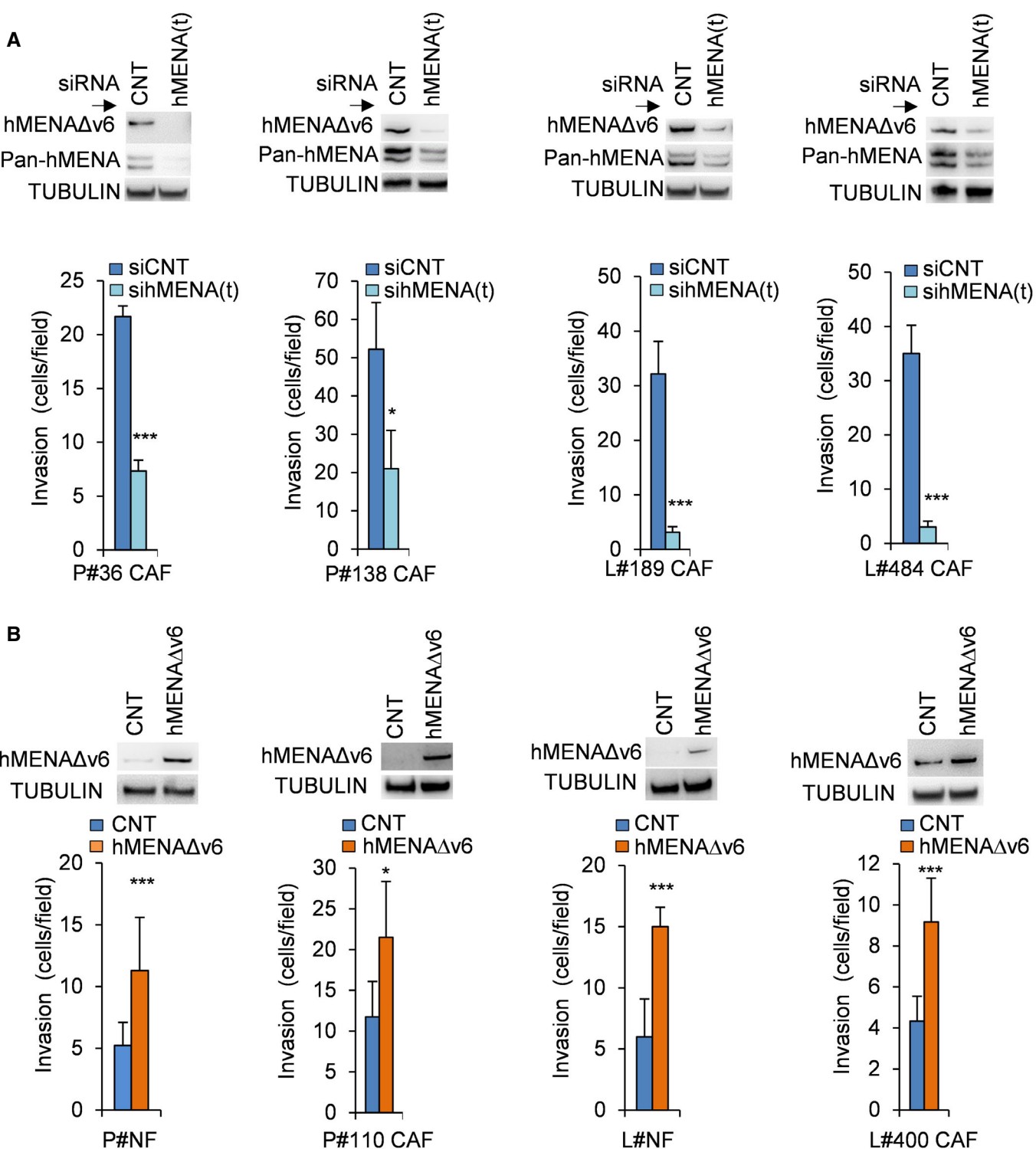

**Figure 3.**

**Figure 3.  hMENA/hMENAΔv6 regulate pro-tumor CAF functional activity.**

A  Quantification of *in vitro* Matrigel invasion assay (bottom) of P-CAF and L-CAF (P-CAF # 36, 138 and L-CAF #189, 484) transfected with control siRNA (CNT) or hMENA siRNA (hMENA(t)) indicating that the siRNA-mediated knock-down of hMENA/hMENAΔv6 reduces the invasive ability of CAFs with respect to siCNT CAFs. Number of invading cells was measured by counting 6 random fields. Data are presented as the mean ± SD of two biological replicates, performed in triplicates each. Immunoblot showing hMENA/hMENAΔv6 expression (detected by Pan-hMENA mAb and by the specific anti-hMENAΔv6 antibody) of the CAFs employed is reported (top). TUBULIN was used as loading control. *P* values were calculated by two-sided Student's *t*-test. *$P < 0.05$, ***$P < 0.001$.

B  Quantification of *in vitro* Matrigel invasion assay (bottom) of P-NF and L-NF and P-CAF#110 and L-CAF#400 transfected with control or hMENAΔv6 expressing vectors, demonstrating that the overexpression of hMENAΔv6 isoform induced the invasiveness of P-NFs and L-NFs and/or P-and L-CAFs. Number of invading cells was measured by counting 6 random fields. Data are presented as the mean ± SD of two biological replicates, performed in triplicates each. Immunoblot of hMENAΔv6 expression (detected by the specific anti-hMENAΔv6 antibody) in fibroblasts employed is reported (top). TUBULIN was used as loading control. *P* values were calculated by two-sided Student's *t*-test. *$P < 0.05$, ***$P < 0.001$.

Source data are available online for this figure.

To identify whether hMENA/hMENAΔv6 expression affects paracrine pro-invasive pathways, we collected conditioned medium (CM) from P-NFs and P-CAFs. According to hMENAΔv6 expression, evaluated in WB analysis, these were classified in P-CAF [high] (hMENAΔv6 expression greater than 2-fold of the average expression in NFs) and P-CAF [low] (hMENAΔv6 expression lower than 2-fold; Fig 1B). We evaluated CM effects on PANC-1 cell invasion, and we found that when PANC-1 were treated with the P-CAF-CM for 48 h an increase of cancer cell invasion was associated with high hMENAΔv6 expression. Indeed, as shown in Fig 4A, CM derived from P-CAFs [high] have a higher pro-invasive effect compared to CM derived from P-CAF [low] and/or from NFs (Fig 4A) indicating that capability to increase cancer cell invasion of CAF-CM is associated with hMENA/hMENAΔv6 expression. In agreement, when CAFs were silenced for hMENA/hMENAΔv6, their CM fails to induce PANC-1 and KP4 PDAC cell invasion (Fig 4B and Appendix Fig S7A). These data were confirmed in H1975 (Fig 4C and D) and A549 NSCLC cells (Appendix Fig S9C). In addition, the CM of silenced P-CAFs also showed a reduced ability to induce *in vitro* tumor cell growth (Appendix Fig S8).

These results indicated that hMENA/hMENAΔv6 overexpression identifies a subset of CAFs with pro-tumor functions, able to regulate tumor cell growth and invasion through the modulation of paracrine factors.

Based on the relevance of bi-directional communication between cancer cells and CAFs, we tested whether the hMENA-mediated CAF activation is reciprocally sustained by hMENA overexpression in tumor cells. We first treated P-NFs with CM derived from PANC-1 cells (highly expressing the hMENA/hMENAΔv6 isoforms) for 24 h, and we assessed the expression of hMENAΔv6 in P-NFs and P-CAF#110. We found that tumor cell-derived secreted factors significantly up-regulated hMENAΔv6, suggesting that tumor-derived secretome is able to induce hMENAΔv6 overexpression in NFs (Fig 4E). This also occurs in CAF with low hMENAΔv6 (P-CAF#110) (Fig 4G) and with high hMENAΔv6 level (L-CAF#484) (Appendix Fig S7B).

To learn in depth whether hMENA/hMENAΔv6 in tumor cells support CAF activation, CAFs were mono-cultured or indirectly co-cultured in a transwell format with control PANC-1 cells (siCNT PANC-1) and/or with PANC-1 cells hMENA/hMENAΔv6 silenced (sihMENA(t) PANC-1). After 48 h, we observed that co-culture of P-CAFs with control tumor cells (siCNT PANC-1) increased CAF gel contraction ability compared to mono-cultured P-CAFs (Fig 4F). Of relevance, the silencing of hMENA/hMENAΔv6 expression in PANC-1 inhibits CAF activation as indicated by the reduced gel

contraction ability of CAFs primed with CM derived from PANC-1 silenced cells (Fig 4F). We also found that the CM derived from PANC-1 cells (control) induced hMENAΔv6 in CAFs (#110 low hMENAΔv6) along with α-SMA expression, and this effect was abrogated when we silenced hMENA/hMENAΔv6 (sihMENA(t) PANC-1) (Fig 4G). These data indicate that hMENA orchestrates the reciprocal interaction between CAF and tumor cells through paracrine factor modulation.

## hMENA/hMENAΔv6 expression in CAFs regulates cancer cell invasion via GAS6

To identify secreted proteins that might account for the pro-tumor functional differences observed between CAF [high] and CAF [low], we performed a LC-MS/MS proteomic analysis on CM derived from P-CAF [high], P-CAF [low], and P-NFs.

We identified, 142, 321 and 387 proteins in CM-NFs, CM-CAF [low], and CM-CAF [high], respectively.

Among the proteins identified in all CM samples, 102 proteins were in common, 25 proteins were up-regulated in CM-CAF [high] compared to CM-NFs, and 16 proteins were up-regulated in CM-CAF [high] compared to CM-CAF [low]. Interestingly, 10 proteins (i.e., SYNC, CACNA1A, EPHA3, MUC16, SIK3, CDH13, GAS6, MYH9, ZBBF2, and CCD37) were uniquely secreted by CM-CAF [high] and defined the hMENAΔv6 associated signature (Fig 5A).

Among the 10 proteins identified, we focused on growth arrest-specific protein 6 (GAS6) considering the main role of its receptor AXL in the mechanisms of drug resistance, mainly mediated by the EMT, including resistance to ICB (Hugo *et al*, 2016; Pallocca *et al*, 2019).

We validated by ELISA and qRT–PCR (Fig 5B and C) that GAS6 is highly secreted by P-CAFs [high] compared to P-CAFs [low] and P-NFs. Differently, we found that PANC-1 cells express low level of GAS6 (Fig 5D), in agreement with previous data indicating the role of GAS6 as stromal-derived factor involved in pro-tumor paracrine-mediated communication (Tjwa *et al*, 2008).

We also found (GSE22862 data set; Navab *et al*, 2011) a significant increase in the expression of GAS6 in L-CAFs compared with L-DFs ($P = 0.0208$) (Fig 5E), in agreement with our LC-MS/MS proteomic analysis in PDAC fibroblasts.

We then explored the role of hMENA/hMENAΔv6 in GAS6 regulation by analyzing the mRNA expression in both P-CAFs and L-CAFs, silenced for hMENA/hMENAΔv6 expression. As shown in Fig 5F,G hMENA/hMENAΔv6 silencing significantly reduced the mRNA expression of *GAS6* in both P-CAFs and L-CAFs. On the other

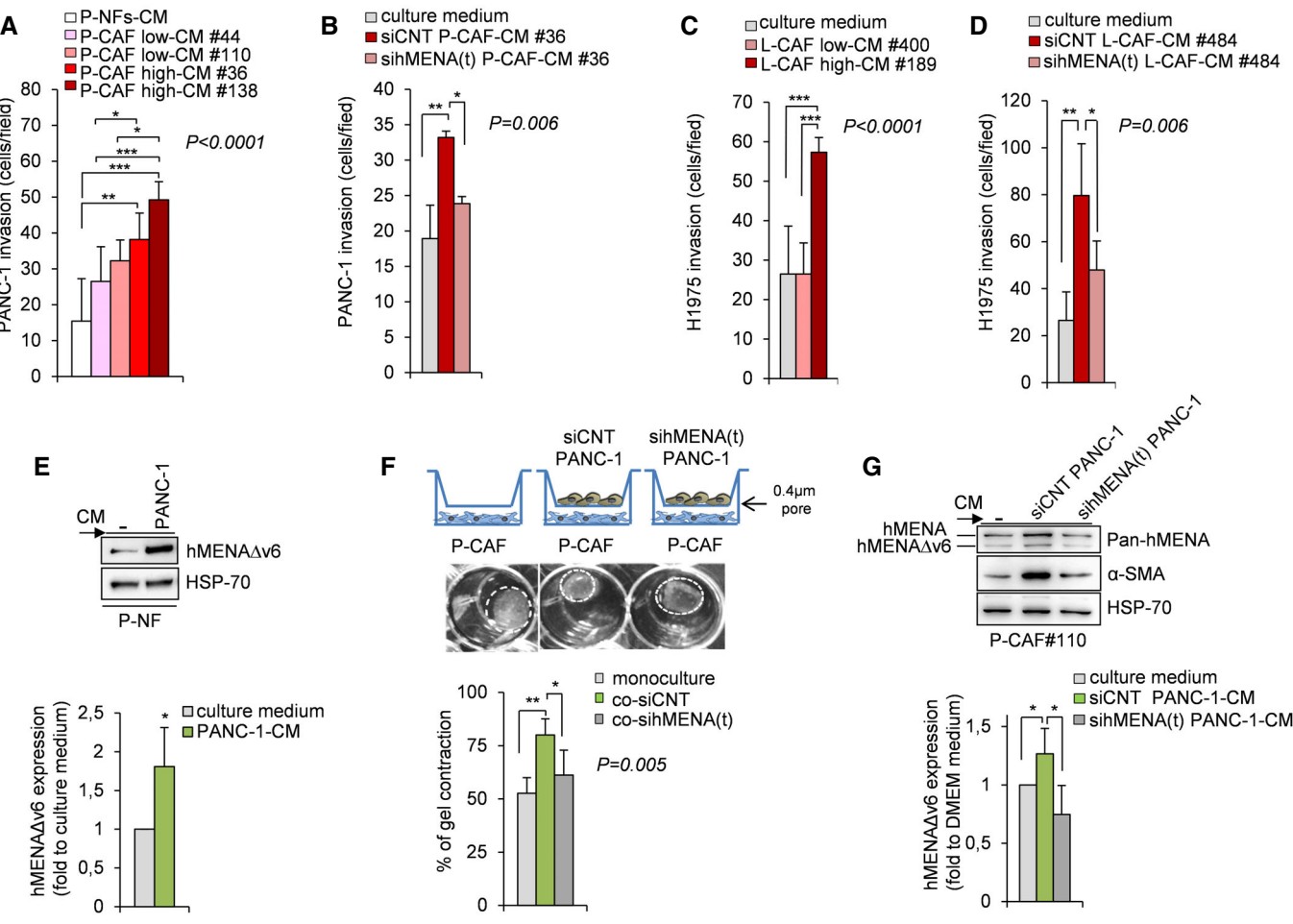

**Figure 4. hMENA/hMENAΔv6 mediates the reciprocal dialogue between tumor cells and CAFs.**

A  Quantification of *in vitro* Matrigel invasion assay of PANC-1 cells cultured for 48 h with conditioned media (CM) of NFs (P-NFs-CM), CAF low #44 and #110 and CAFs high #36 and 138. Histograms show the number of invading cells measured by counting 6 random fields. Data are presented as the mean ± SD of three biological replicates, performed at least in duplicate each. Statistical analysis was performed with one-way ANOVA P < 0.0001, followed by Bonferroni's multiple comparison test. *P < 0.05, **P < 0.01, ***P < 0.001.

B  Quantification of *in vitro* Matrigel invasion assay of PANC-1 cultured for 48 h with CM derived from control P-CAFs#36 (siCNT-P-CAF-CM#36) and hMENA/hMENAΔv6 silenced P-CAFs (sihMENA(t)-P-CAF-CM#36), showing that the siRNA-mediated knock-down of hMENA/hMENAΔv6 affects PANC-1 invasive ability mediated by CAF-CM. Culture medium (DMEM) was used as control. Cells invading Matrigel were counted in 6 random fields. Data are presented as the mean ± SD of three biological replicates. Statistical analysis was performed with one-way ANOVA P = 0.006, followed by Bonferroni's multiple comparison test. *P < 0.05, **P < 0.01.

C  Quantification of *in vitro* Matrigel invasion assay of H1975 cells cultured for 48 h with control media (culture medium) or conditioned media (CM) of L-CAF low #400 and CAFs high #189, as described above. Data are presented as the mean ± SD of two biological replicates, performed in triplicates each. Statistical analysis was performed with one-way ANOVA P < 0.0001, followed by Bonferroni's multiple comparison test. ***P < 0.001.

D  Quantification of *in vitro* Matrigel invasion assay of H1975 cultured for 48 h with CM derived from control L-CAFs#484 (siCNT-L-CAF-CM#484) and hMENA/hMENAΔv6 silenced L-CAFs (sihMENA(t)-L-CAF-CM#484), showing that the siRNA-mediated knock-down of hMENA/hMENAΔv6 affects H1975 invasive ability mediated by CAF-CM. Culture medium (DMEM) was used as control. Cells invading Matrigel were counted in 6 random fields. Data are presented as the mean ± SD of six replicates. Statistical analysis was performed with one-way ANOVA P = 0.006, followed by Bonferroni's multiple comparison test. *P < 0.05, **P < 0.01.

E  Representative immunoblot (top) and densitometry quantification (bottom) of hMENAΔv6 expression level in P-NFs grown in RPMI control medium (−) or PANC-1−CM for 24 h (n = 3). Data are represented as fold increase with respect to control medium ± SD (set as 1). Data were analyzed using two-sided Student's *t*-test. *P < 0.05.

F  Representative images (top) and quantification (bottom) of collagen gel contraction ability of CAFs (monoculture) or CAFs co-cultured with siCNT (co-siCNT) or sihMENA(t) PANC-1 cells (co-sihMENA(t)). Dashed white circles illustrate the margins of the gel area. Data are presented as the mean ± SD of two biological replicates. Statistical analysis was performed with one-way ANOVA P = 0.005, followed by Bonferroni's multiple comparison test ANOVA: *P < 0.05, **P < 0.01.

G  Representative immunoblot (top) and quantification (bottom) of hMENAΔv6 and α-SMA expression in P-CAFs #110 treated with DMEM (−) or with conditioned medium (CM) derived from siCNT (siCNT PANC-1) or sihMENA(t) PANC-1 cells ((sihMENA(t) PANC-1) (n = 3). Data are represented as fold increase with respect to DMEM (set as 1 ± SD). Data presented were analyzed using two-sided Student's *t*-test: *P < 0.05.

Source data are available online for this figure.

hand, the silencing of *GAS6* does not affect hMENA/hMENAΔv6 expression as shown in Appendix Fig S9B.

Further, we asked whether the effect of hMENA/hMENAΔv6 in CAF-driven tumor cell invasion relies on the ability of hMENA to

regulate GAS6 secretion. To this end, we treated PANC-1 for 48 h with CM derived from P-CAFs silenced for GAS6 expression (Appendix Fig S9A), and we observed that GAS6 silencing in CAFs reduced tumor cell invasion (Fig 5H), as hMENA/hMENAΔv6

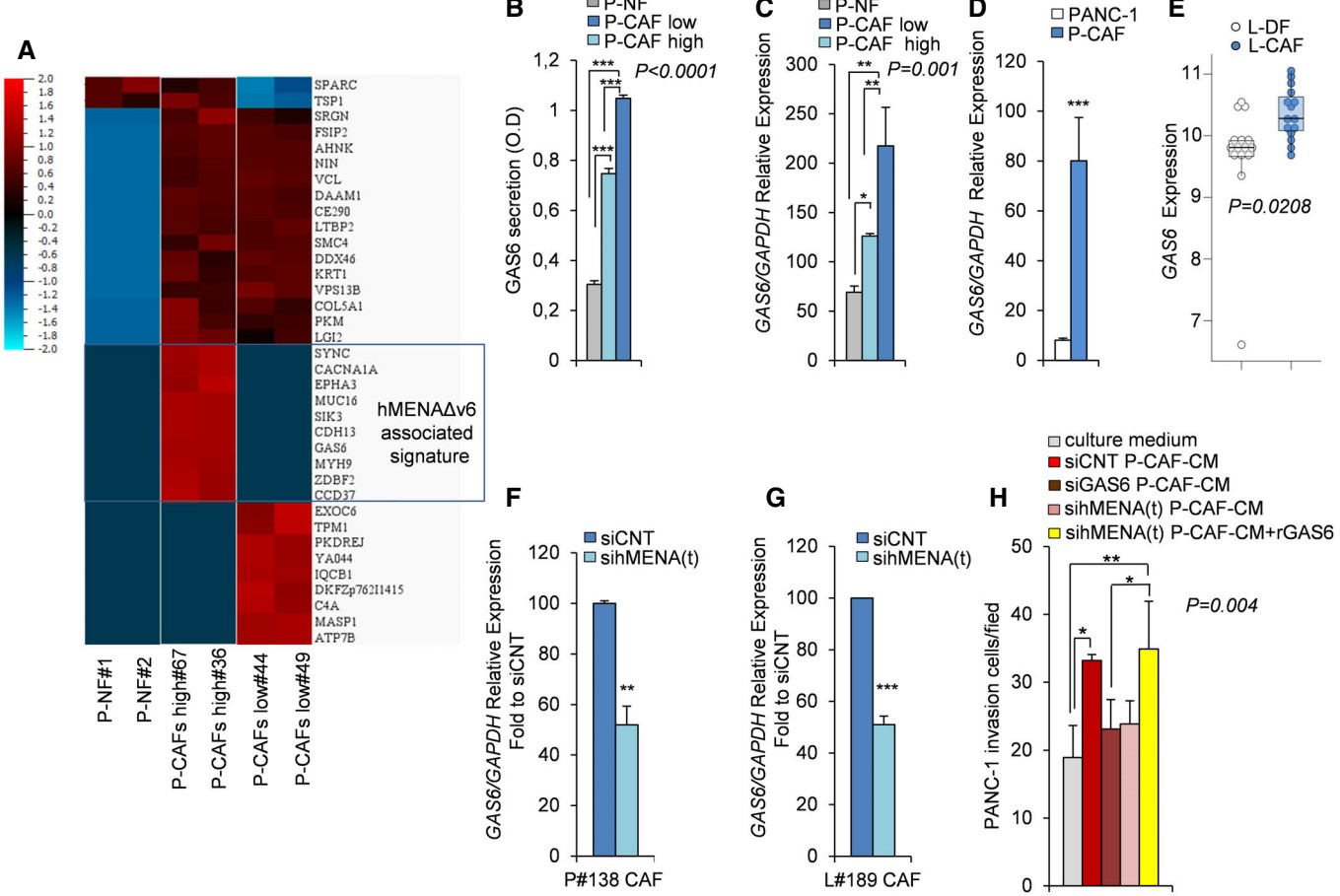

**Figure 5. High hMENAΔv6 CAFs secrete GAS6 required for cancer cell invasion.**

A    Heat map of proteins identified by LC-MS/MS analysis as differentially secreted in CM of normal fibroblasts (P-NF#1 and P-NF#2), CAFs with low level of hMENAΔv6 expression (P-CAFs low#44 and #49), CAFs with high level of hMENAΔv6 expression (P-CAFs high #67 and #36).
The boxed hMENAΔv6 associated signature represents proteins exclusively present in the CM derived from CAFs ^high^. Color code is shown in the upper left corner. Co-variance = 0.2, *q*-value = 0.1 (10% false discovery rate), adjusted *P* value = 0.0074542. *N* = 36. The values of mass spec identified for each protein were plotted in log₂ scale.

B    Quantification of GAS6 secretion levels, as detected by ELISA, in the CM of P-NF (#1), P-CAF^low^ (#44 and #49) and P-CAF ^high^ (#67 and #36). Data are presented as the mean ± SD of two biological replicates. Statistical analysis was performed with one-way ANOVA *P* < 0.0001, followed by Bonferroni's multiple comparison test. ***P* < 0.001.

C    Real-time qRT–PCR analysis of the relative GAS6 mRNA expression level in NFs (P-NF#1), P-CAF ^low^ and P-CAF ^high^, as described above. Data are presented as the mean ± SD of three replicates. Statistical analysis was performed with one-way ANOVA *P* = 0.001, followed by Bonferroni's multiple comparison test. **P* < 0.05, ***P* < 0.01.

D    Real-time qRT–PCR analysis of the relative GAS6 mRNA expression level in PANC-1 and P-CAF (*n* = 3), showing a low GAS6 expression in PANC-1 cells. Data are presented as the mean ± SD. *P* values were calculated by two-sided Student's *t*-test. ****P* < 0.001.

E    Boxplots showing the mRNA expression of GAS6 in normal lung fibroblasts (white) (*n* = 15) versus primary NSCLC fibroblasts (light blue) (*n* = 15) (GSE22862 data set). In each boxplot the median value (horizontal line), 25^th^–75^th^ percentiles (box outline), and highest and lowest values within 1.5× of the interquartile range (vertical line) are shown. Statistical significance was calculated by Mann–Whitney *U*-test (two-sided) (*P* = 0.0208).

F, G    Real-time qRT–PCR analysis of P#138 CAF (F) and L#189 CAF (G) transfected with control siRNA (siCNT) or hMENA(t) siRNA (sihMENA(t)) as representative cases. The siRNA-mediated knock-down of hMENA/hMENAΔv6 resulted in a significant reduction of GAS6 mRNA expression levels compared to siCNT cells (set as 100). Data are presented as the mean ± SD of three replicates. *P* values were calculated by two-sided Student's *t*-test ***P* < 0.01, ****P* < 0.001.

H    Quantification of Matrigel invasion assay of PANC-1 cultured for 48 h with DMEM (culture medium), or conditioned medium (CM) derived from control siRNA P#106 CAFs (siCNT P-CAF-CM), GAS6 siRNA (siGAS6 P-CAF-CM), hMENA(t) siRNA (sihMENA(t) P-CAF-CM), hMENA(t) siRNA plus rGAS6 (sihMENA(t) P-CAF-CM + rGAS6). Number of invaded PANC-1 cells after 48 h of treatment was measured by counting 6 random fields. Data are presented as the mean ± SD of three biological replicates. Statistical analysis was performed with one-way ANOVA *P* = 0.004, followed by Bonferroni's multiple comparison test. **P* < 0.05, ***P* < 0.01.

silencing in CAFs did (Fig 4B and D). This pro-invasive CAF-derived GAS6 was also obtained in L-CAFs (Appendix Fig S9C).

Importantly, the addition of recombinant GAS6 (rGAS6) to hMENA(t) silenced CAFs rescued cancer cell invasiveness in PANC-1 (Fig 5H). These data demonstrate that hMENA/hMENAΔv6 regulate GAS6 expression and secretion in CAFs and in turn CAF-mediated cancer cell invasiveness.

### hMENA/hMENAΔv6 regulate AXL expression in tumor cells and sustain the paracrine GAS6-AXL-mediated tumor cell/CAF pro-invasive cooperativity

The GAS6-dependent activation of the receptor tyrosine kinase AXL has been shown to increase the invadopodia functions (Revach et al, 2019). We have recently reported that hMENA/hMENAΔv6 exert a pivotal function in ET-1/β-arr1-induced invadopodial activity and ovarian cancer invasiveness (Di Modugno et al, 2018a).

To identify a role of hMENA in tumor and CAF cells pro-invasive cooperativity, we firstly examined whether hMENA regulates AXL receptor expression in tumor cells. We found that the silencing of hMENA (sihMENA(t)) reduces AXL protein levels in PANC-1 (Fig 6A) and KP4 PDAC (Appendix Fig S10) and in A549 (Fig 6A) H1650, H1975 NSCLC cells (Appendix Fig S10). This reduction occurred also at mRNA level as evaluated in PANC-1 and A549 cells (Fig 6B). To assess the effect of hMENA silencing on AXL gene transcription, we measured both total AXL mRNA and AXL pre-mRNA by qRT–PCR from total RNA isolated from siCNT and sihMENA(t) PANC-1 and A549 cell lines at 72 h post-siRNA transfections. As shown in Fig 6C, both pre-mRNA and mature mRNA AXL levels were decreased in hMENA-silenced cells, suggesting that hMENA regulates AXL at transcription level (Fig 6C). The reduction of sensitivity of hMENA(t) silenced cancer cells to the AXL expression-dependent BGB324 kinase inhibitor (R428) further confirmed that hMENA silencing induced AXL downregulation (Appendix Fig S11A). However, BGB324 did not affect hMENA expression as evaluated in a panel of cancer cell lines (Appendix Fig S11B).

To investigate the role of hMENA in ligand-dependent AXL signaling, we treated PANC-1 cells with rGAS6 and we found that hMENA silencing inhibits GAS6-mediated AXL and AKT phosphorylation (Fig 6D) and reduced cancer cell invasion toward GAS6 as detected by Matrigel transwell invasion assay (Fig 6E). Accordingly, these data have been also validated in the NSCLC cell line H1975 (Appendix Fig S7C). Notably, according to the pro-invasive role of CAF-derived GAS6 on PANC-1 (Fig 5H), tumor cells silenced for hMENA were less invasive when treated with CAF-CM (Fig 6F). These data indicate that hMENA exerts its pro-invasive role playing a crucial function in the communication between cancer cells and CAFs via the regulation of the GAS6/AXL paracrine axis.

### Combined expression of hMENA, AXL, and GAS6 is a gene signature associated with a poor outcome in pancreatic adenocarcinoma and lung squamous carcinoma patients

To define the relevance of our experimental data in clinical practice, we investigated the prognostic value of the combined ENAH (hMENA), AXL, and GAS6 mRNA expression levels in pancreatic cancer (PDAC) patients (n = 172), lung squamous carcinoma (LUSC) (n = 501), and lung adenocarcinoma (LUAD) patient

subtypes (n = 516) from The Cancer Genome Atlas (TCGA) (Network, 2012), (Liu et al, 2018).

Interestingly, the 3-gene (ENAH, AXL, and GAS6) expression signature correlated with a worse prognosis for overall survival (OS) in PDAC, stratifying the patients on the basis of the signature expression (HR = 1.97, 95% CI: 1.24–3.13, P = 0.0034; Fig 7A left). In contrast, no prognostic correlation was found neither with the 1-gene signature (ENAH) (HR = 1.28, 95% CI: 0.81–2.02, P = 0.29; Fig 7A right) or with the 2-gene (AXL and GAS6) expression signature (HR = 1.33, 95% CI: 0.83–2.14, P = 0.24; Fig 7A middle). Notably, a correlation between elevated 3-gene signature expression, but not 1 or 2-gene signature expression, and shorter disease-specific survival (DSS) was also found (HR = 1.74, 95% CI: 1.08–2.81, P = 0.012; HR = 1.30, 95% CI: 0.82–2.08 P = 0.47 and HR = 1.26, 95% CI: 0.77–2.06, P = 0.114, respectively), underlining the clinical relevance of these three genes in PDAC (Appendix Fig S12), and indicating that concomitant expression of ENAH with AXL and GAS6 confers a prognostic value to this gene signature.

A similar prognostic correlation for the 3-gene expression signature (HR = 1.596, 95% CI: 1.13–2.25, P = 0.0069) (Fig 7B left) but not for the 1 or 2-gene expression signature (HR = 1.32, 95% CI: 0.95–1.83 P = 0.23 and HR = 1.32, 95% CI: 0.95–1.83, P = 0.095, respectively; Fig 7B middle and right) was found in LUSC patients (OS at 80 months).

Collectively, these results establish the 3-gene (ENAH, AXL and GAS6) expression signature as a prognostic indicator, hallmark of an aggressive disease in PDAC and LUSC patients and strengthen the clinical relevance of the hMENA expression pattern analysis in both tumor cells and CAFs.

## Discussion

Tumor evolution is shaped by the reciprocal interaction between cancer and non-cancerous organ-specific cells (Bissell & Radisky, 2001). Many studies have pointed on the main role of cancer cells/ CAF interaction and relative autocrine and paracrine signaling they activate (Tape et al, 2016; Sahai et al, 2020). However, only few factors have been implied as able to drive malignant cell program and CAF cells reprogramming reciprocally (Scherz-Shouval et al, 2014), with important therapeutic implication.

Herein we propose that the inhibition of the actin cytoskeleton regulatory protein hMENA may interrupt the communication between cancer cells and CAFs leading to an inhibition of tumor invasiveness by regulating GAS6/AXL axis and impacts PDAC and NSCLC patient prognosis.

By isolating CAFs from a large number of PDAC and NSCLC primary tissues, we found a novel role for the actin cytoskeleton regulatory protein hMENA and its mesenchymal tissue-specific isoform hMENAΔv6 in CAF activation, highly complementary to their previously defined functions in tumor cells (Di Modugno et al, 2012).

As novelty in support of our previous data that hMENA/ hMENAΔv6 participate at invadopodia maturation and mediate cancer cell invasiveness (Di Modugno et al, 2012, 2018a), herein we demonstrate that hMENA regulates the expression of the RTK AXL recently reported to be involved in the regulation of invadopodia formation (Revach et al, 2019). Reciprocally, we found that

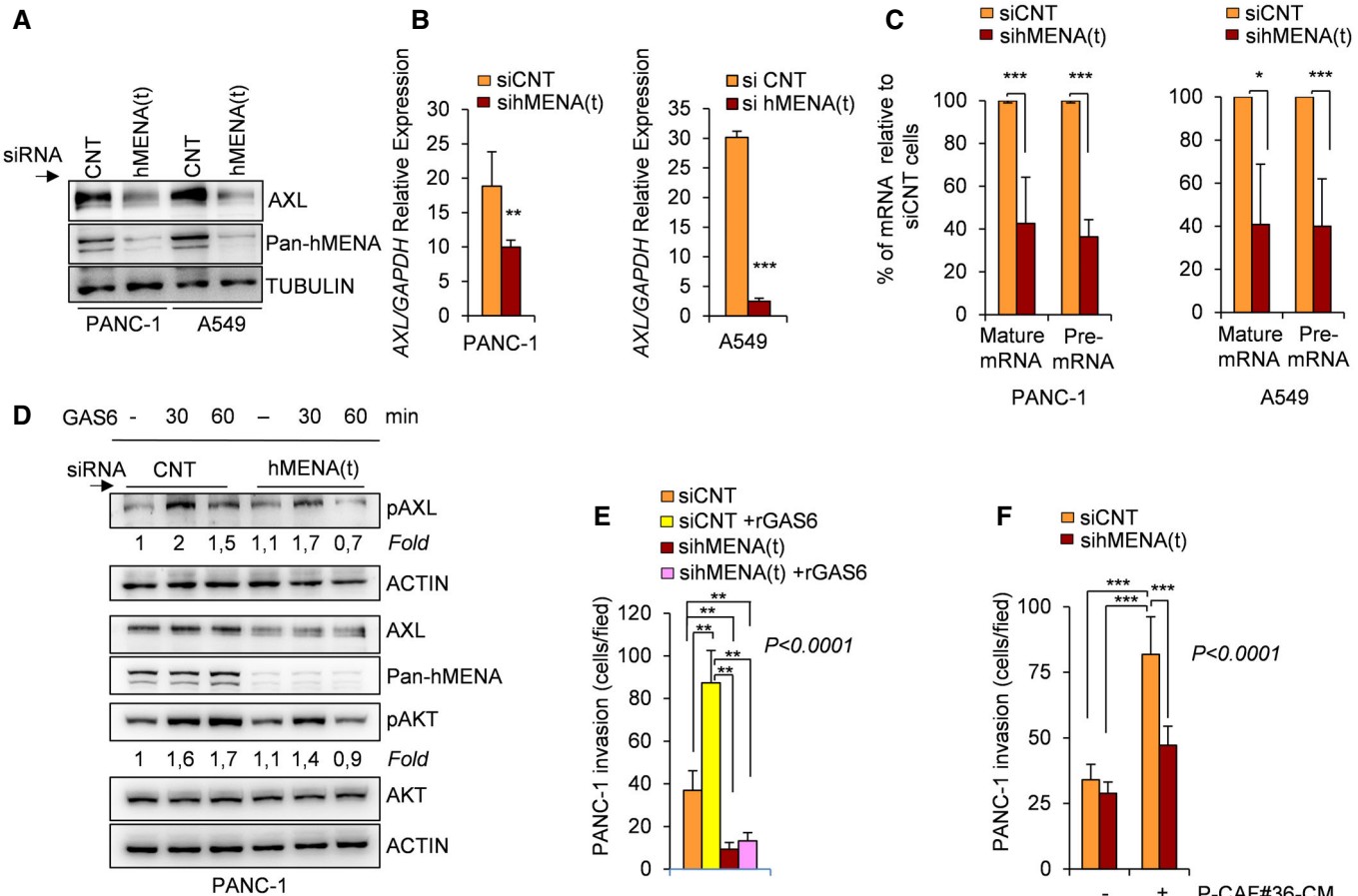

**Figure 6. hMENA/hMENAΔv6 silencing inhibits AXL expression and activity in cancer cells.**

A   Immunoblot of AXL expression in PANC-1 and A549 cancer cells upon transfection of control siRNA (CNT) or hMENA(t) siRNA (sihMENA(t)), indicating that the knock-down of hMENA isoforms, resulted in a reduction of AXL protein expression.

B   Real-time qRT–PCR analysis of the relative AXL mRNA expression in PANC-1 and A549 cancer cells transfected with control siRNA (siCNT) or hMENA(t) siRNA (sihMENA(t)), indicating that the knock-down of hMENA isoforms, resulted in a significant reduction of mRNA AXL expression. Data are presented as the mean ± SD of three replicates. P values were calculated by two-sided Student's t-test. **P < 0.01, ***P < 0.001.

C   Real-time qRT–PCR analysis of the relative levels of mature AXL mRNA or AXL pre-mRNA in PANC-1 (left) and A549 (right) cell lines, transfected with control siRNA (siCNT) or hMENA(t) siRNA, sihMENA(t). Data represent percent of AXL mRNA or pre-mRNA levels in hMENA(t) silenced cells relative to siCNT control cells (set at 100). Data are presented as the mean ± SD of two biological replicates. P values were calculated by two-sided Student's t-test. *P < 0.05, ***P < 0.001.

D   Immunoblot analysis with the indicated antibodies of PANC-1 cells transfected with control siRNA (CNT) or hMENA(t) siRNA, showing that the knock-down of total hMENA isoforms, (hMENA(t)) inhibits GAS6-mediated pAXL and pAKT expression. Cells were serum starved overnight and subsequently stimulated with DMSO (0.02%) in control culture medium (−) or rGAS6 (200 ng/ml), for 30 and 60 min. The fold change of pAXL or pAKT expression respect to siCNT untreated cells is reported.

E   Quantification of in vitro Matrigel invasion assay of PANC-1 siCNT cells (siCNT) and hMENA/hMENAΔv6 silenced cells (sihMENA(t)) toward rGAS6 as chemo-attractant (siCNT + rGAS6 and sihMENA(t) + rGAS6), showing that the knock-down of hMENA(t) reduced cancer cell invasion toward GAS6. The number of invading cells was counted in 6 random fields. Data are presented as the mean ± SD of three biological replicates. Statistical analysis was performed with one-way ANOVA P < 0.0001, followed by Bonferroni's multiple comparison test. **P < 0.01.

F   Quantification of in vitro Matrigel invasion assay of PANC-1 siCNT cells (siCNT) and hMENA/hMENAΔv6 silenced cells (sihMENA(t)), untreated (−) or treated with conditioned media derived from CAFs (P-CAF#36-CM) for 48 h. The number of invaded cells was counted in 6 random fields. Data are presented as the mean ± SD of three biological replicates. Statistical analysis was performed with one-way ANOVA P < 0.0001, followed by Bonferroni's multiple comparison test. ***P < 0.001.

Source data are available online for this figure.

hMENA/hMENAΔv6 regulate the AXL ligand GAS6 expression and secretion in CAFs, indicating hMENA as a crucial player in the tumor invasiveness mediated by the cooperativity between cancer cells and CAFs.

Based on our expression and functional data, we indicate that hMENA/hMENAΔv6 is expressed at very low level in normal fibroblasts and it is highly expressed in activated CAFs (Figs 1B and C, and 3 and Appendix Fig S6). Our observation that in primary PDAC

and NSCLC tissue hMENA/hMENAΔv6 were expressed in CAF (Fig 2 A and B) was confirmed by the data that in a comprehensive catalog of stromal cells described at single cells resolution (Lambrechts et al, 2018; Peng et al, 2019), hMENA is expressed only in CAFs and not in immune cells (Fig 2C and D). However, hMENA is differently expressed in the different CAF "clusters" identified in the Lambrechts' study and our characterization of PDAC and NSCLC CAFs clearly evidenced this heterogeneity of expression

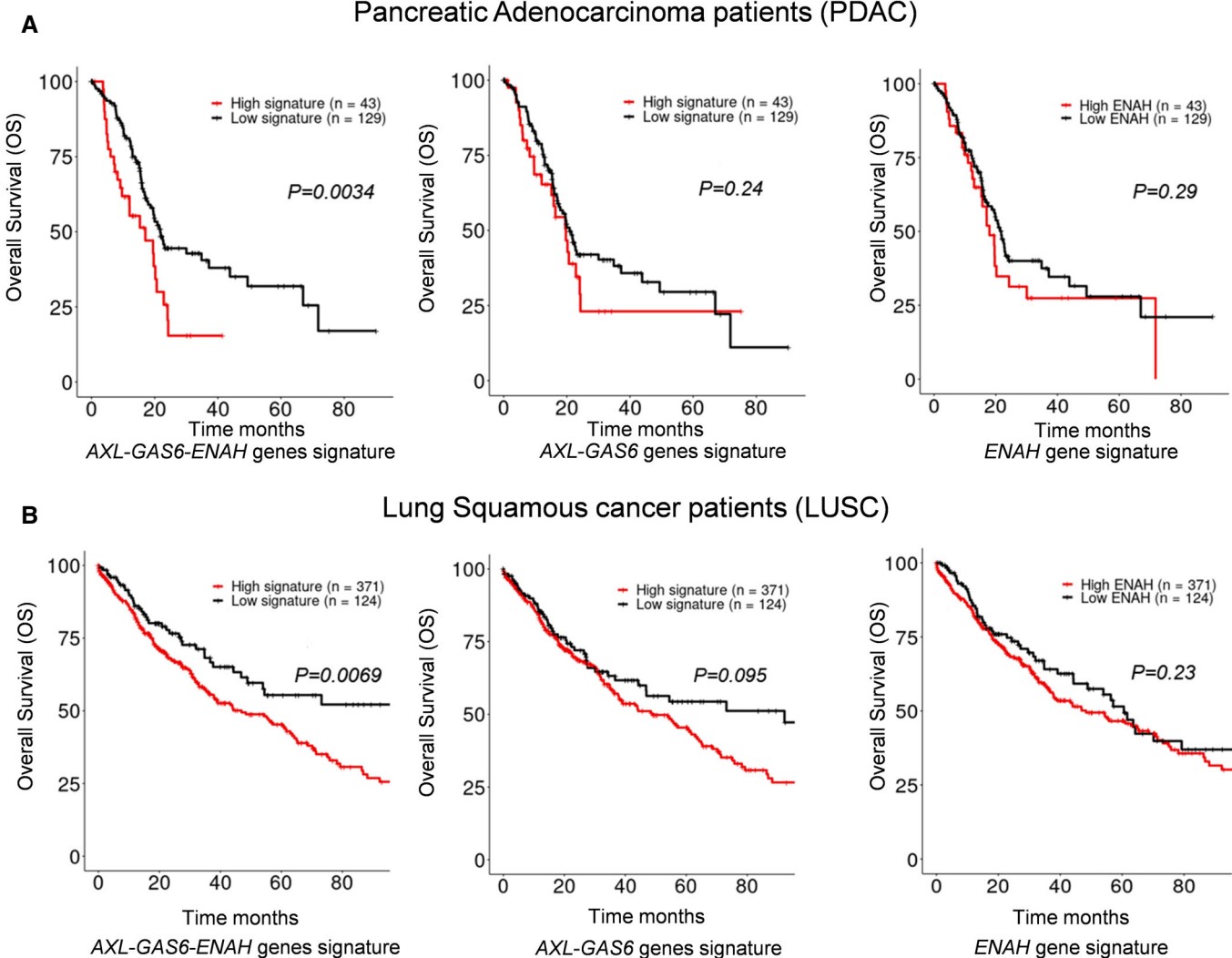

**Figure 7. Combined expression of hMENA, AXL, and GAS6 correlated with decreased survival in PDAC and NSCLC cancers.**

A  Overall survival (OS) curves in pancreatic adenocarcinoma patients (PDAC) (n = 172) from The Cancer Genome Atlas (TCGA) showed that combined expression of AXL, GAS6, and ENAH (hMENA) is a prognostic signature in PDAC.

B  Overall survival (OS) curves in lung squamous cancer patients (LUSC) (n = 501) from The Cancer Genome Atlas (TCGA) showed that combined expression of AXL, GAS6, and ENAH (hMENA) is a prognostic signature in LUSC.

Data information: In (A, B) patients were stratified in three groups on the basis of the AXL/GAS6/ENAH (left), AXL/GAS6 (middle), and ENAH (right) signature expression levels. P values are shown. Statistical significance was calculated by using the log-rank test.

(Appendix Fig S3B). Four different CAF subsets identified by analyzing specific CAF markers (FAP, integrin β1/CD29, αSMA, S100-A4/FSP1, PDGFRβ, and CAV1) have been reported and related to immune modulation in human breast and ovarian cancers (Costa et al, 2018; Givel et al, 2018).

Noteworthy, distinct population of CAFs named as "myCAFs", "iCAFs", and "apCAFs" with the ability to dynamically reverse from one cell state to the other have been described in PDAC, suggesting that pancreatic CAF subpopulations represent dynamic and interconvertible states (Öhlund et al, 2017).

We were unable to cluster hMENA-expressing CAFs in known CAF subtypes and, by analyzing the "Navab data sets" (Navab et al, 2011), we found that CAF subgroup with high hMENA expression

showed an increased α-SMA and FAP expression compared with the CAFs with low hMENA expression (hMENA low). Considering the main role of hMENA in actin dynamic regulation, we argue that hMENA rather than identify a specific CAF subtype mirrors a CAF state reflecting the pro-invasive functions. Indeed, we demonstrated that the silencing of hMENA/hMENAΔv6 reduces pro-tumor CAF functionality. Of relevance, conversely the overexpression of the hMENAΔv6 isoform in normal fibroblasts leads to CAF activation (Fig 3B and Appendix Fig S6) and in turn to cancer cell invasiveness (Fig 4A–D).

We hypothesized that in the secretome of hMENA/hMENAΔv6 high CAFs prevail pro-invasive factors and by LC-MS/MS proteomic analysis, we identified a series of proteins (i.e., SYNC, CACNA1A,

EPHA3, MUC16, SIK3, CDH13, GAS6, MyH9, ZBBF2, and CCD37) defining the hMENAΔv6-associated signature (Fig 5A).

GAS6 is a crucial factor involved in paracrine stroma-tumor cell interaction, as evidenced by several studies (Tjwa *et al*, 2008), and the GAS6-AXL axis has been implicated in the reciprocal signaling between PDAC and stromal cells induced by KRASG12D mutation (Tape *et al*, 2016). Thus, we in depth analyzed whether hMENA actively participate in the GAS6-AXL pathway regulation. Our interest is also dictated by the relevance of this axis in drug resistance, mainly mediate by the EMT, immune suppression (Scaltriti *et al*, 2016), and by the recent indication to combine AXL targeting compounds with immune checkpoint inhibitors (Akalu *et al*, 2017). Our data, that the silencing of hMENA/hMENAΔv6 in CAFs reduced GAS6 expression and secretion leading to the inhibition of CAF-induced cancer cell invasiveness (Fig 5B, C, F, G, H), are in agreement with a recent study showing that CAF-derived GAS6 can activate AXL and promote tumor cell migration (Kanzaki *et al*, 2017). Of biological and clinical relevance herein, we report that hMENA/hMENAΔv6 play a dual role in the regulation of GAS6/AXL axis. Indeed, hMENA expression not only promotes the secretion of GAS6 in CAFs but also regulates AXL expression and GAS6-mediated AXL activation in tumor cells (Fig 6A–E). It is worth mention that we have previously shown that hMENAΔv6 regulates vimentin expression at RNA and protein level (Melchionna *et al*, 2016), and induction of vimentin by EMT has been associated with upregulation of AXL expression (Vuoriluoto *et al*, 2011).

Although the exact role of hMENA in the regulation of AXL expression remains to be fully elucidated, we have found that hMENA silencing decreased the level of both total and AXL pre-mRNA expression suggesting a contribution of hMENA to AXL transcription (Fig 6C).

Furthermore, two main hypotheses can be argued based on our previous data that hMENA abundance, by regulating F-/G-actin ratio (Di Modugno *et al*, 2018b), may modify cell tension (Tavares *et al*, 2017); (i) hMENA modulates nuclear translocation of co-factors and transcription factor activation as we have recently reported for MRTF-A and SRF (Di Modugno *et al*, 2018b) and among the genes regulated by SRF; AXL has listed as one of the targets genes from ChIP-seq data available in the Harmonizome tool as revealed by querying the targetome of SRF (Rouillard *et al*, 2016); (ii) hMENA determines change in protein conformation and phosphorylation states as we have previously reported for various RTK such as epithelial growth factor receptor (EGFR), HER2, and HER3 (Di Modugno *et al*, 2007, 2010).

Functionally we found that the downregulation of hMENA/hMENAΔv6 expression in tumor cells inhibits the GAS6-induced cancer cells invasiveness, indicating that hMENA expression in both tumor cell and CAFs may empowered the paracrine GAS6-AXL axis.

Clinically we found that, in a large cohort of PDAC and NSCLC clinical specimens, the integrated high expression levels of ENAH/AXL/GAS6 positively correlate with poor prognosis (Fig 7A and B; Appendix Fig S12) providing a novel predictive signature in PDAC and NSCLC progression.

From a therapeutic perspective, our findings that hMENA is able to regulate the pro-tumor CAF activation and their reciprocal interaction with tumor cells make this protein and its tissue-specific splicing isoforms attractive targets for cancer therapy.

# Materials and Methods

## Patients main characteristics and isolation of human cancer-associated and normal fibroblasts

Cancer-associated fibroblasts were obtained from fresh PDAC (P-CAF) and NCSLC (L-CAF) tumor specimens of patients undergoing curative surgery at the Regina Elena National Cancer Institute during years 2012–2018. Patients' main characteristics are listed in Appendix Table S1 and S2. The study and the informed consent obtained from enrolled patients was reviewed and approved by the local ethics committee (Protocol CE/594/11 on 11/03/2011 and 058.IFO_AOO.REGISTRO UFFICIALE.U.0012817.20-11-2017).

Tumor tissues were washed three times in PBS, then minced into approximately 1–2 mm$^2$ sized pieces and digested with 10 ml of 0.1% trypsin (Gibco, Invitrogen). The homogenate (isolated cells and partially digested tissues) was centrifuged at 100 $g$ for 5 min at room temperature (RT) and cultured in Dulbecco's modified Eagle medium (DMEM Gibco, Invitrogen) supplemented with 1% (vol/vol) penicillin/streptomycin, 1% glutamine and 5% (vol/vol) inactivated fetal bovine serum at 37°C in 5% $CO_2$ 95% air. After 24 h, the medium was replaced to eliminate floating cells.

Distal fibroblasts (L-DFs) derived from "non-tumoral" tissue isolated at least 5 cm away from the tumor core were obtained using the same protocol. Normal fibroblasts (P-NF) obtained from normal pancreatic tissues derived from transplant donors were kindly provided by Dr Piemonti, IRCCS San Raffaele Scientific Institute, Milan. All the fibroblasts used in the experiments were at passage between three and seven.

## Sequencing on Ion S5

DNA and RNA used for parallel analysis of NGS and PCR were extracted by All Prep DNA/RNA mini kit (80204; Qiagen). 10 ng of purified genomic DNA was used for library construction with the Oncomine™ Solid Tumour DNA Kit (Life Technologies) that targets regions of human somatic variants (deletions, insertions, inversions, and substitutions) on the following 22 cancer-related genes: EGFR, ALK, ERBB2, ERBB4, FGFR1, FGFR2, FGFR3, MET, DDR2, KRAS, PIK3CA, BRAF, AKT1, PTEN, NRAS, MAP2K1, STK11, NOTCH1, CTNNB1, SMAD4, FBXW7, and TP53, for analysis using Ion Torrent next-generation sequencing technology. Libraries were prepared from 10 ng of DNA extracted from CAFs by using the Ion Ampliseq™ Kit for Chef DL8 (Thermo Fisher Scientific) and Oncomine™ Solid Tumour panel according to the manufacturer's instructions. Libraries were re-loaded into the Ion Chef™ instrument (Thermo Fisher) for emulsion PCR, and templates were prepared using Ion 510™ & Ion 520™ & Ion 530™ Kit-Chef (Thermo Fisher) and sequenced on Ion S5™ System.

## Data analysis

Data analysis, including alignment to the hg19 human reference genome and variant calling, was done using the Torrent Suite Software v.5.10.2 (Thermo Fisher). Variants were identified by Ion Reporter filter (software v.5.4 Thermo Fisher). The limit of detection for variant calling was > 5%. Filtered variants were annotated using Ion Reporter software v.5.4 (Thermo Fisher).

## CAF cancer cells co-cultures and conditioned media preparation

Co-culture is based on the combined culture of 1:1 mix of PANC-1 and P-CAFs in transwell format (BD Biosciences 0.4 μm pore size). Conditioned medium (CM) was prepared from CAFs or tumor cells grown in 35 mm culture plates in serum-free media (1 ml) for 48 h. CM were collected, centrifuged (450 $g$, 4 min), and incubated with cancer cells for the indicated time points.

## Cell lines and cell culture treatments

The human PDAC and NCSLC cell lines PANC-1, A549, H1650, H1975, CALU-1 were purchased from American Type Culture Collection and cultured in RPMI 1640 medium (Gibco, Invitrogen) supplemented with 1% (vol/vol) glutamine and 10% (vol/vol) inactivated fetal bovine serum at 37°C in 5% $CO_2$ 95% air. Normal lung fibroblast (L#NF) IMR-90 was from ATCC and cultured in EMEM (Gibco, Invitrogen) with 1% (vol/vol) glutamine and 10% (vol/vol) inactivated fetal bovine serum. PDAC KP4 cells were provided by the cell culture facility at The University of Texas MD Anderson Cancer Center (UTMDACC; Houston, Texas) and were maintained in Iscove's modified Dulbecco's medium (Gibco), supplemented with 20% FBS and 200 mmol/l L-glutamine. Normal dermal fibroblasts (HNF) were kindly provided by Silvia Soddu (Regina Elena National Cancer Institute, Rome).

Recombinant GAS6 (885-GS; R&D Systems) was added to cell media of PANC-1 cells at 200 ng/ml, according to the effect evidenced by a dose curve.

The selective inhibitor of AXL R428 (BGB324) was purchased from MedChem Express (HY-15150) and added to cell media at 1 and 2.5 μM.

Cancer cell lines were periodically authenticated by BMR Genomics. All cell lines were routinely checked for Mycoplasma using Mycoplasma PCR Reagent set (Euroclone).

## Immunofluorescence staining

Cancer-associated fibroblasts were cultured on μ-slide 8-well glass bottom (IBIDI), precoated with gelatin 0.2% and grown for 48 h before fixing, permeabilizing as previously reported (Di Modugno et al, 2012) and stained with: anti-α-SMA Mouse antibody (1:100; 1A4; Dako), anti-pan cytokeratin Mouse antibody (1:50; AE2/AE3; Novocastra, Leica Microsystems), anti-Pan-hMENA Mouse antibody (1: 1,000; A351F/D9; Millipore), anti-hMENA[11a] Mouse antibody (1:1,000; Di Modugno et al, 2012), anti-E-Cadherin Mouse antibody (1:200; clone 36; BD Biosciences), anti-FAP antibody (1:100; ab28244; Abcam). After multiple washes with PBS, the cells were incubated with anti-mouse (Alexa Fluor 555) or anti-rabbit (Alexa Fluor 488) secondary antibodies (1:250; Thermo Fisher Scientific) for 30 min at 37°C. Nuclei were counterstained with DAPI (Bio-Rad).

Serial 3-μm histological sections from paraffin-embedded surgical sections were deparaffinized in xylene and hydrated in descending dilutions of ethanol and then washed in $H_2O$. Slides were placed in a water bath at 95°C for 40 min in the citrate buffer (pH 6.0; S2031; Dako) for antigen retrieval. The slides were incubated at room temperature (RT) in glycine 0.1 M in PBS for 20 min, in 0,1% Triton X-100 in PBS for 5 min and then in blocking buffer with 5% BSA (sc-2323; Santa Cruz Biotechnology), 5% FBS in PBS, for 30 min; followed by overnight at 4°C with anti-Pan-hMENA Mouse antibody (1: 200; A351F/D9; Millipore) and anti-α-SMA rabbit antibody (1:200; ab5694; Abcam) in blocking buffer. The Alexa Fluor 594-conjugated goat anti-rabbit IgG antibody or Alexa Fluor 647-conjugated goat anti-mouse antibody (1:250; Thermo Fisher Scientific) were diluted in blocking buffer and used to incubate the slides for 1 h at 37°C. Nuclei were counterstained with DAPI (Bio-Rad).

Immunofluorescence was analyzed by Zeiss LSM 880 with Airy scan confocal laser scanning microscope equipped with a 20× air or 63X/1.23 NA oil immersion objectives. Lasers 405, 488, 514, 633 nm were used to excite the fluorophores. The Zeiss Zen control software (Zeiss, Germany) was used for image analysis.

## Transfection, and Small interfering RNA (siRNA)

Cancer-associated fibroblasts were transiently transfected with hMENAΔv6 cDNA or with vector alone (pcDNA3), using Lipofectamine 3000 (Invitrogen) according to the manufacturer's instructions. For the small interfering RNA, cells in exponential growth phase were transfected with 20 nmol/l of hMENA(t)-specific pooled siRNA duplexes (siGENOME SMARTpool Human ENAH), 2 nmol/l of GAS6-specific pooled siRNA duplexes (ON-TARGETplus SMARTpool Human GAS6-2621), or 20 nmol/l of ON-TARGETplus Non-targeting Control Pool (GE Healthcare, Dharmacon,) using Lipofectamine® RNAiMAX Transfection Reagent (Invitrogen) according to the manufacturer's protocol. hMENA knock-down was also obtained by transient transfection of MISSION® shRNA Plasmid DNA-ENAH human-TRCN0000303614 (Sigma-Aldrich) (Di Modugno et al, 2018b). The effects of silencing were evaluated at 48–72 h from the transfection by Western blot analysis.

## RNA analysis

Total RNA (5 μg) was isolated using trizol reagent (Invitrogen) and transcribed into cDNA by first-strand cDNA synthesis kit (27-9261-01; GE Healthcare,) according to the manufacturer's protocol. Platinum Pfx DNA polymerase (11708013; Invitrogen) was used for semi-quantitative RT–PCR reactions, and the inclusion or skipping of exons 11a and 6 was analyzed by using hMENA-specific primers, as already reported (Di Modugno et al, 2012) P7 forward 5'-GA ATTGCTGAAAAGGGATCGAATTGCTGAAAAGGGATC-3' and P8 reverse 5' CTGTTCCTCTATGCAGTATTTGAC-3' flanking the exon 11a to detect the inclusion/skipping of exon 11a or with primers MTC1 forward 5'-GCTGGAATGGGAGAGAGAGCGCAGAATATC-3' and MTC2 reverse 5'-GTTCACACCAATAGCATTCCCTCCACTTG-3' flanking exon 6. RT–PCR for β-actin was performed as control of normalization, and PCR products were run on 1% agarose gels. Quantitative real-time RT–PCR (qRT–PCR) was performed on an ABI Prism 7500 Real-time PCR instrument (Applied Biosystems). The reactions were carried out in triplicates, and the primer sequences used for the qRT–PCR were as follows: AXL forward 5'-CTCGCCGCCTCCGTACATGTC-3'; AXL reverse 5'-GCCCCGCCTTAT GATTCTCTGC-3'; GAS6 forward 5'-CTGGATGG-TGGTGTCTTCTC-3'; GAS6 reverse 5'-GACCTGCCAAGACATAGACG-3; GAPDH forward 5'-TCCCTGAGCTGAACGGGAAG-3'GAPDH reverse 5'-GGAGGAGT GGGTGTCGCTGT -3'.

TaqMan Gene Expression Assays were used for amplification and quantification of ENAH (Hs00430216; Applied Biosystems), EPCAM (Hs00901885_m1; Applied Biosystems), PDGFRB (Hs01019589_m1; Applied Biosystems), and of human hypoxanthine phosphoribosyltransferase 1 gene (HPRT1)(4331182; Applied Biosystems), used as an endogenous control. The comparative Ct method ($2-\Delta\Delta/C_t$ method) was used to determine changes in relative levels of different genes (Livak & Schmittgen, 2001). For the intronic PCR, total RNA was extracted from A549 and PANC-1 hMENA-silenced and control cells using the MasterPure RNA Purification Kit (Epicentre Biotechnologies). To remove genomic DNA contamination from RNA samples, DNAse I treatment was performed. RNA was reverse-transcribed into cDNA using the High-Capacity cDNA Reverse Transcription Kit (4368813; Applied Biosystems) and subject to StepOne Real-Time PCR (4376357; Applied Biosystems) with PowerUp SYBR Green Master Mix (A25779; Applied Biosystems). A reaction using the standard amplification mix was done in the absence of reverse transcriptases (-RT) to control the RNA template for DNA contamination. Primer pairs for PCR amplification of both intronic and exonic sequences were designed as specified below: Intron #1 forward 5′-CCAGGCAGTG AATTTGGGTG-3′; Intron #1 reverse 5′-ACAGAGTCCTTGATGCGA TCC-3′; Intron #2 forward 5′-GAATCAGAATGAGGGCAAGGG-3′; Intron #2 reverse 5′-AGTTGAGCAAGCACCATCTCA-3′; Exon #1 forward 5′-CCAGCACCTGTGGTCATCTT-3′; Exon #1 reverse 5′-CAC ATTGTCACCCCGAAGGA-3′; Exon #2 forward 5′-CTGAGTGAAGCG GTCTGCAT-3′; Exon #2 reverse 5′-CATCTTGGCGATACGTCCCT-3′. PCR amplification conditions were as follows: 50°C for 2 min, 95°C for 2 min, followed by 40 cycles of 95°C/15 s, annealing at 56°C/ 30 s and 72°C/30 s. All reactions were performed in triplicate. Data were normalized to GAPDH, and the change in gene expression relative to siCNT cells (set as 100%) was calculated using the comparative Ct method (Livak & Schmittgen, 2001).

## Western blot analysis

Protein extraction and Western blot analyses were carried out as previously described (Di Modugno *et al*, 2012). Protein concentrations were determined by bicinchoninic acid (BCA) assay (#23225; Pierce) according to the manufacturer's instructions. The following antibodies were used in this study: anti-hMENAΔv6 Rabbit antibody (1:2,500; Di Modugno *et al*, 2012), anti-Pan-hMENA Rabbit antibody (1:5,000; Di Modugno *et al*, 2012), hMENA[11a] Mouse antibody (1:2,000; Di Modugno *et al*, 2012), anti-AXL Rabbit antibody (1:1,000; C89E7; Cell Signaling Technology), anti-phospho-AXL (Y779) Rabbit antibody (1:1,000; AF2228; Cell Signaling), anti-α−SMA Rabbit antibody (1:1,000; Cell Signaling Technology), anti-AKT Rabbit antibody (1:1,000; 11E7; Cell Signaling Technology), anti-phospho-AKT (ser473) Rabbit antibody (1:1,000; D9E; Cell Signaling Technology), anti-FAP Rabbit antibody (1:500; ab28244; Abcam), anti-β-actin Mouse antibody (1:2,500; A4700; Sigma-Aldrich,), anti-HSP70 Mouse antibody (1:2,000; sc-24; Santa Cruz Biotechnology), anti-α-Tubulin Rabbit antibody (1:1,000; 11H10; Cell Signaling Technology), and the secondary antibodies goat anti-Rabbit HRP (1:2,000; 170-6515; Bio-Rad) and goat anti-Mouse HRP (1:2,000; 170-6516; Bio-Rad). Densitometric quantitation of antibodies immunoreactivity was determined by ImageJ software (https://imagej.nih.gov/ij/) and

normalized in comparison with the β-actin, tubulin, and/or HSP70 immunoreactivity.

## Immunohistochemistry

Pan-hMENA (mouse, clone 21, 610693, BD Biosciences) and immunoreactions were revealed by Bond Polymer Refine Detection (Leica Biosystem, Milan, Italy) on an automated autostainer (BondTM Max, Leica) as previously described in ref. Di Modugno *et al* (2012).

## Collagen gel contraction assay

Fibroblasts (100,000 cells) were resuspended in 200 μl of collagen 1 solution (2 mg/ml). The collagen solution (1 ml) was prepared by mixing 0.8 ml of PureCol Bovine Collagen I type (Roche), 0.2 ml of 5× DMEM (Gibco), and 40 μl of 7.5% $NaHCO_3$, pH = 8.

Cell suspension was cast into a single well of a 24-well tissue culture plate. After polymerization, the collagen gels were released from the surface of the culture well using a sterile tip. Gel contraction was evaluated at 24 h and quantified by taking photographs (iPhone 6). The relative area of the well and the gel were measured using ImageJ 1.49v program (NIH), and the percentage of contraction was calculated using the formula:

$$100 \times (\text{well area} - \text{gel area})/\text{well area}$$

The ability of CAF co-cultured with cancer cells to contract collagen gel was measured after 48 h of co-culture as described above.

## ELISA assay

Levels of GAS6 in conditioned media were determined by ELISA (DY885B human GAS6 DuoSet ELISA kit). The culture medium of NFs, CAFs[low], and CAFs[high] was collected, and GAS6 level was detected according to the manufacturer's protocol (R&D Systems CA, USA).

## Gelatin zymography

Conditioned medium of silenced and/or transfected cells was collected after 48 h from transfection. MMP-2 activation was analyzed by gelatin zymography using 10% polyacrylamide resolving gel containing 1 mg/ml gelatine. After electrophoresis, gels were washed with 50 mM Tris–HCl pH 7.5, 5 mM $CaCl_2$, and 2.5% Triton X-100 and then incubated in 50 mM Tris–HCl pH 7.5, 5 mM $CaCl_2$ at 37°C overnight. Gels were stained with 0.25% Coomassie brilliant blue (R-250) dye in 10% acetic acid and 10% isopropanol. Semiquantitative densitometry was performed using the ImageJ software (https://imagej.nih.gov/ij/).

## Cell invasion assay

$5 \times 10^5$ cells/ml were seeded in Matrigel invasion chamber (8.0 μm pore size; BD Biosciences, 354480). Top chambers (culture inserts) were filled with serum-free medium, and bottom chambers were filled with medium containing 10% FBS. After

24 h, cells at the upper membrane surface were removed with a cotton swab. Cells on the lower side of the filters were fixed, stained with Diff quick (Sigma-Aldrich), photographed, and at least 6 fields were counted. Each experiment was performed for three times, in triplicates. For CAF-induced cancer cell invasion, cancer cells were pre-treated or not with CM derived from control and/or hMENA(t) silenced CAF for 48 h, and then, invasion assays were performed as described above. For Matrigel invasion assay toward GAS6, recombinant GAS6 (200 ng/ml) were added into the lower chamber. Each group was plated in duplicates in each experiment, and the experiment was repeated at least 2 times.

### Crystal violet cell growth assays

Cells were seeded into 24-well plates (in triplicates per condition) and treated with CAF-derived CM for 72 h. Following treatments, cells were fixed for 10 min in 4% paraformaldehyde and stained with in 0.1% crystal violet solution (Sigma). The dye was subsequently extracted with 10% acetic acid, its absorbance was determined (570 nm).

### Secretome analysis of CAFs by liquid chromatography coupled with tandem mass spectrometry (LC-MS/MS)

For MS analysis, CAF were grown in serum-free DMEM and CM collected after 48 h of culture; protease and proteinase inhibitors were added immediately. CM was centrifuged at 2,500 $g$ for 30 min to remove cell debris, and the supernatant was concentrated using the Amicon Ultra 15, Ultracel-3K centrifugation device (Thermo Fisher) until the volume was reduced to ~100 μl. The protein concentration was determined using the Qubit Protein Assay (Thermo Fisher).

### Secretome analysis of CAFs by liquid chromatography coupled with tandem mass spectrometry (LC-MS/MS)

In Solution Protein Digestion: 5 μg of proteins were reduced, alkylated, and digested in 50 μl of 200 ng Trypsin Gold reconstituted in 50 mM ammonium bicarbonate at 37°C overnight. After the digestion was complete, the peptide mix was centrifuged subsequently for 30 min at 16,000 $g$, and the cleared supernatants were transferred to fresh tubes to be dried and resuspended in 0.1% TFA for subsequent peptide fractionation using the Pierce High pH Reversed-Phase Peptide Fractionation Kit (Thermo Fisher). Peptide fractionations were collected for LC-MS/MS analysis. LC-MS/MS analysis: The dried peptide mix was reconstituted in a solution of 2% acetonitrile (ACN), 2% formic acid (FA) for MS analysis. Peptides were loaded directly onto a 2 cm C18 PepMap pre-column by an autosampler (Thermo Scientific), which was coupled to a 50 cm EASY-Spray C18 analytical column (Thermo Scientific). Peptides were eluted from the column using a Dionex Ultimate 3000 Nano LC system with a 2 min gradient from 2% buffer B to 5% buffer B (100% acetonitrile, 0.1% formic acid), followed by a 65 min gradient from 5% buffer B to 20% buffer B and a 15 min gradient from 20% to 30% buffer B. The gradient was switched from 30% to 85% buffer B over 1 min and held constant for 3 min. Finally, the gradient was changed from 85% buffer B to 98% buffer A (100% water, 0.1% formic acid) over 1 min and then held constant at 98% buffer A for 20 more minutes. The application of a 2.0 kV distal voltage electrosprayed the eluting peptides directly into the Thermo Fusion Tribrid mass spectrometer equipped with an EASY-Spray source (Thermo Scientific). Mass spectrometer-scanning functions and HPLC gradients were controlled by the Xcalibur data system (Thermo Finnigan). MS data were acquired in the Fourier Transforming (FT) at 120,000 resolution from m/z 400 to 1,600. CID MS/MS were acquired in the IT on 2+ and higher charge state ions for 3 s duty cycles.

### Database search and interpretation of MS/MS data

Acquired MS data were searched against a human UniProt database (released on 4/27/2016 and common contaminants were added) using Proteome Discoverer 1.4 with fixed modifications of carbamidomethyl on cysteine and possible oxidation on methionine. The Proteome Discoverer probability-based scoring system rates the relevance of the best matches found by the SEQUEST algorithm. The peptide mass search tolerance was set to 10 ppm. A minimum sequence length of 7 amino acids residues was required. Only fully tryptic peptides were considered. To calculate confidence levels and FDR, Proteome Discoverer generates a decoy database containing reverse sequences of the non-decoy protein database and performs the search against this concatenated database (non-decoy + decoy). Scaffold (Proteome Software) was used to visualize searched results. The discriminant score was set to be less than 1% FDR, which are determined based on the number of accepted decoy database proteins to generate protein lists for this study. Spectral counts were used for semi-quantitative comparisons of protein abundance among samples.

### Survival analysis on TCGA datasets

Expression and survival data for the analysis of The Cancer Genome Atlas (TCGA), pancreatic cancer (PDAC), lung squamous carcinoma (LUSC), and lung adenocarcinoma (LUAD) program were from the Pan-Cancer Atlas publication (Liu et al, 2018) and downloaded from the Genomic Data Common repository (https://gdc.cancer.gov/node/905/). Expression values from the same sample but from different vials/portions/analytes/aliquots were averaged.

For overall survival (OS) and disease-specific survival (DSS) analysis, patients with available specific survival data were considered. The gene signature expression levels were performed considering the average of the z-score scaled expressions of the genes in the signature. Patients were stratified into two groups on the basis of the signature expression levels using quartiles as thresholds, and the best fits are reported. In detail, for PDAC patients were reported the curves using as cut-off the 75th percentile of the signature expression and for LUSC patients were reported the curves using as cut-off the 25th percentile of the signature expression. The curves were estimated using the Kaplan–Meier method, and the differences were tested using the log-rank test. $P$ values < 0.05 were considered statistically significant. All the analyses were performed with R software (version 3.5.3) and the survival package.

### Single-cell RNA-Seq data analysis on pancreatic ductal adenocarcinoma dataset

Single-cell RNA-Seq row data from 24 pancreatic ductal adenocarcinoma (PDAC) tumor samples were from the Peking Union Medical College Hospital and downloaded from the Genome Sequence Archive (project PRJCA001063; Peng *et al*, 2019).

The paired-end FASTQ files were processed by the Cell Ranger Single-Cell Software Suite (version 3.1.0) pipeline with standard parameters that performed the alignment on reference transcriptome (GRCh38-3.0.0), filtering, and unique molecular identifier (UMI) counting (cellranger count command). The resulting gene-barcode matrix containing the barcoded cells, and gene expression counts was then imported into the Seurat (v. 3.1.1) R toolkit (Satija *et al*, 2015) to perform the quality control and downstream analysis as indicated in the Peng *et al* paper. Firstly, low-quality cells (< 200 genes/cell, < 3 cells/gene and > 10% mitochondrial genes) were excluded. Next, the dimensionality reduction was performed using the principal component analysis (PCA) and the statistically significant principal components were identified using the jackstraw approach with 100 iterations (JackStraw and ScoreJackStraw functions). Principal components 1–10 were used for graph-based clustering (with resolution parameter set to 0.5) to identify distinct groups of cells. To present data in two-dimensional coordinates, t-distributed stochastic neighbor embedding (t-SNE) implemented in RunTSNE function was used. Finally, the identities of cell types of these groups were characterized according to the expression of known markers as indicated in the Peng *et al* paper. In detail, 10 groups of cells were identified: acinar (414 cells), B cells (1,734 cells), ductal 1 (2,164 cells), ductal 2 (10,973 cells), endocrine (326 cells), endothelial (4,637 cells), fibroblast (5,889 cells), macrophage (4,656 cells), stellate (4,579 cells), and T cells (1,891 cells). 1,224 cells were not distinctively characterized. For each patient, the mean expression of ENAH in each group of cells was computed and plotted. 2 patients (T2 and T7) resulted to have only "not distinctively characterized" cells and were removed.

### Statistical analysis

Statistical significance for two-sample comparisons was calculated by two-tailed unpaired Student's *t*-test (two-sided) or Mann–Whitney *U*-test (two-sided) depending on the distribution of the data, as indicated; statistical significance for multiple-sample comparisons was calculated by one-way ANOVA. Where more than one test was performed, *P* values were corrected for multiple testing using the Benjamini–Hochberg (BH) method. The difference was considered statistically significant if *P* value, corrected in case of multiple test, is lower than 0.05. All data are presented as mean ± standard deviation (SD) of at least three independent experiments, unless otherwise stated.

Statistical tests used to assess significance of differences between means are indicated in each Figure legend. *P* values below 0.05 were marked by *, while **$P < 0.01$ and ***$P < 0.001$. *P* values < 0.05 were considered significant. All the statistical analyses were performed with R software (version 3.5.3) and SPSS software (version 21.0).

## Data availability

Mass spectrometry data have been deposited to the Mass Spectrometry Interactive Virtual Environment (MassIVE) mass spectrometry data repository and can be accessed using the following link: https://massive.ucsd.edu/ProteoSAFe/static/massive.jsp with the access ID: MSV000084685.

**Expanded View** for this article is available online.

### Acknowledgements
We thank all patients who donated samples for this study. We thank Giuliana Falasca, Maria Vincenza Sarcone and Vittoria Balzano for technical assistance. This work was supported by the Italian Association for Cancer Research AIRC: 5 × 1,000, 12182 (P.N., A.S.) and IG 19822 (P.N.). Synopsis image was created with BioRender.com (http://biorender.com/).

### Author contributions
RM and PN designed and wrote the study. RM performed most of the experiments with SS. ADC and AMM contributed to WB analysis and intronic PCR, respectively; FDM provided guidance and feedback on the experimental design, prepared the figures, and revised the manuscript; DDA analyzed the TGCA data sets and performed bioinformatics analysis; IS performed statistical analysis; MP provided technical support; BA performed immunohistochemistry; EG performed NGS analysis; PV provided pathological review of patients; GLG and FF surgically managed PDAC and NSCLC patients and provided clinical data; EC conducted the LC-MS/MS proteomics experiments and interpreted the relative results; LP provided normal fibroblasts; AS, MM, and RTL provided expert feedback. AS reviewed the manuscript; PN conceived and directed the study. All authors critically read this manuscript.

### Conflict of interest
The authors declare that they have no conflict of interest.

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
