## [Review Process File · EMBO Reports]

The actin modulator hMENA regulates GAS6-AXL axis and pro-tumor cancer/stromal cell cooperation

Roberta Melchionna, Sheila Spada, Francesca Di Modugno, Daniel D'Andrea, Anna Di Carlo, Mariangela Panetta, Anna Maria Mileo, Isabella Sperduti, Barbara Antoniani, Enzo Gallo, Rita T Lawlor, Lorenzo Piemonti, Paolo Visca, Michele Milella, Gianluca Grazi, Francesco Facciolo, Emily Chen, Aldo Scarpa, and Paola Nistico

DOI: [10.15252/embr.202051077](https://doi.org/10.15252/embr.202051077)

Corresponding author(s): Paola Nistico (paola.nistico@ifo.gov.it)

Review Timeline:

Submission Date:	21st Jan 20
Editorial Decision:	24th Feb 20
Revision Received:	17th Jun 20
Editorial Decision:	31st Jul 20
Revision Received:	4th Aug 20
Accepted:	10th Aug 20

Editor: Deniz Senyilmaz Tiebe

Transaction Report:

Dear Dr. Nisticò,

Thank you for submitting your manuscript entitled 'The actin modulator hMENA regulates Gas6-Axl axis and cooperativity between cancer and stromal cells' to EMBO Reports. We have now received three referee reports, which are included below.

Referees express interest in the proposed role of hMENA in regulation of GAS6/AXL axis NSCLC and PDAC. However, referees, especially 1 and 2, point out a number of concerns regarding the conclusiveness of the dataset and the methodologies used, therefore do not recommend publication here. In particular, the referees raise concerns regarding

1. Insufficient biological replicates
2. Insufficient characterization of CAFs
3. Unclear biological relevance of hMENA levels in the cell lines used for its depletion.

Given these comments from recognized experts in the field that are also experienced reviewers, and considering the amount of work required to address them, we cannot offer to publish your manuscript.

However, in case you feel that you can address the referee concerns in a timely and thorough manner, and can obtain data that would considerably strengthen the study as in the referee reports, we would have no objection to consider a revised manuscript (along with a point-by-point response to the referee concerns) in the future. Please note that if you were to send a new manuscript this would be assessed again with respect to the literature and the novelty of your findings at the time of resubmission and in case of a positive editorial evaluation, the manuscript would be sent back to the original referees. I would like to emphasize that we will be reluctant to approach the referees again in the absence of major revisions, and we need strong support from the referees to consider publication here.

Thank you in any case for the opportunity to consider this manuscript and my apologies once again for this unusual delay in the process. I am sorry that I cannot communicate more positive news, but nevertheless hope that you will find our referees' comments helpful.

Yours sincerely,

Deniz Senyilmaz Tiebe

Deniz Senyilmaz Tiebe, PhD
Editor
EMBO Reports

Referee #1:

Melchionna et al describe a new role for hMENA in regulating the GAS6/AXL axis in pancreatic ductal adenocarcinoma and non-small cell lung cancer by inducing the expression of the ligand GAS6 in cancer-associated fibroblasts and of the GAS6-receptor AXL in cancer cells. The authors perform a number of well-connected experiments and add some novelty to what previously shown.

These results could not only be relevant to PDAC and NSCLC, but also to other malignancies. However, to support their claims, the authors should apply a more rigorous experimental approach, as outlined below. Additionally, a number of concerns need to be addressed prior to consideration for publication in EMBO reports.

Major Points

- 1- The authors should better characterize the CAF lines generated to make sure that they are not EMT cells. Especially considering the previous description of pro-invasive hMENADv6 in E-cadherin-low mesenchymal cells (di Modugno 2012). For example, in addition to alpha-SMA, which can be expressed by EMT cells, the authors should check other fibroblast markers by IF (e.g. FAP, PDGFR). Additionally, the authors should confirm that the CAF lines used do not carry any mutation present in the tumour sample of origin (e.g. KRAS, EGFR). A characterization of bona fide CAFs would strengthen the significance of this study.
- 2- Although we appreciate that the authors show for all experiments at least 2 technical repeats, the majority of experiments are only shown for 1 PDAC CAF line, and for 1 NSCLC CAF line. The authors should show at least 2 biological replicates (i.e. 2 different CAF lines) for all experiments shown.
- 3- Similar to the point above, for all relevant experiments, the authors should use a second PDAC cancer line in addition to Panc-1, and a second NSCLC line in addition to A549 (the authors do have H1975 and H1650 cells, but only use them once).
- 4- We appreciate that the authors have used several CAF lines throughout the paper. However, a major concern is the fact that no single CAF line has been used for all different experiments shown (e.g. in Figure 3, three different lung CAF lines have been used for three different experiments). This lack of consistency is a significant weakness of the paper, as it is difficult to assess whether the conclusions made by the authors are generalizable to various PDAC and NSCLC CAFs or are due to cell line-specific features. To support their claims, the authors should perform all experiments with at least the same 2 PDAC CAF lines and the same 2 lung CAF lines. Inclusion of these data would significantly strengthen the authors' conclusions.
- 5- In Figure 5, the authors use P#110 CAFs and L#182 CAFs for hMENA knock-down experiments. However, in Figures 1D and 1E, the levels of hMENADv6 are very low in these CAF lines. The authors should justify why they chose these CAF lines and explain why they think they are more relevant than using CAF lines with higher hMENA basal levels.
- 6- The authors show that a high HENA/GAS6/AXL signature is associated with poor prognosis in PDAC and NSCLC. However, in their previous publications, as also pointed out in the introduction, they have already shown that hMENA alone is associated with poor prognosis in cancer. The authors should compare the 2 signatures and comment on why it is relevant/novel to point out that a high HENA/GAS6/AXL signature is associated with poor prognosis.
- 7- In Figure 2, the authors identify potential CAFs in tissues due to their elongated morphology. However, this is not sufficient to identify CAFs. To corroborate their point, the authors should perform co-IF with hMENA and FAP or, at least, alpha-SMA. Additionally, the authors should do this co-IF for PDAC tissues, as most of the in vitro data presented focus on PDAC CAF lines.
- 8- The authors should include the knock-down and over-expression validation (qPCR and/or western blot) for all cell lines used, and clarify which lines they show in Figures S4A and S4B. Also, the authors use some CAF lines whose hMENADv6 levels are not shown in Figures 1D and 1E. The authors should include these data as well.
- 9- The authors should include additional relevant references. For example, at the beginning of the introduction, when mentioning the role of the stroma in "hampering treatment efficacy", the authors only cite Provenzano et al, but several other papers should be cited (see David Tuveson, Rakesh Jain, Ronald Evans, etc work). Additionally, at the beginning of the discussion the authors state that "many studies" have pointed out the cancer/CAF signalling but only cite Tape et al. Indeed, many recent studies on PDAC have contributed to this discussion and should be acknowledged (see

Tony Hunter, David Tuveson, Ben Stanger, etc work).

Minor Points

- 1- Please consider the journal's guidelines before finalizing the manuscript (e.g. the title should not include not commonly used abbreviations and it should not be longer than 100 characters; the table legends are missing and should be included).
- 2- Please use correct nomenclature for gene names (i.e. human genes/mRNA should be in capital and italics).
- 3- Please define all acronyms the first time they are introduced (e.g. TGF-beta, EGFR).
- 4- Please make sure that all scale bars are big enough to be clearly visible.
- 5- Please include pan-hMENA blot for Figures 1D and 1E. This is relevant since in some knock-down assays the authors use CAF lines with really low levels of hMENADv6, and it is unclear why using these lines is relevant (see major point #5 above).
- 6- The authors should clarify whether the Student's t test used is unpaired or paired.
- 7- What is the effect of hMENA knock-down and overexpression on the growth and alpha-SMA levels of CAFs? This is relevant as the authors claim that hMENA has a crucial role in CAF activation. Similarly, do P-CAF^{High} proliferate at the same rate of P-CAF^{Low}?
- 8- In Figure 4A, the authors should indicate which CAF line corresponds to which column, as they only use a few lines for their following experiments. The authors should also clarify whether P-NFs corresponds to P-NF1 and P-NF2 or other lines.
- 9- The authors should at least justify why in different CM experiments they use 24h or 48h timepoints (e.g. for Panc1 growth and invasion assays). Similarly, they should clarify why they use transwells in some cases and CM in others. Consistent experimental conditions would allow to compare different assays more appropriately.
- 10- We suspect that Figure S5 (right panel) may have been mistakenly labelled (x axis), and that it refers to Panc-1 cells.
- 11- In Figures S6A and S6B, the authors should clarify which P-CAF line is shown.

Not addressing the following points will not preclude this manuscript from publication, but we find they would be useful for the readership.

- 12- In Figures 1B and 1C, please indicate which hMENA isoform corresponds to which band as done for Supp Fig S6.
- 13- The authors show IF staining for pan-hMENA and hMENA11a. As showing the mesenchymal-specific isoform would be relevant, the authors should also show, if possible, IF and IHC with the hMENADv6 antibody (for Figure 1 and 2). We, however, appreciate that this may not be possible for technical reasons. If so, we would encourage the authors to mention this somewhere in the manuscript.
- 14- In Figure 2C, the authors show increased ENAH expression in fibroblasts compared to other stromal cells. Since the Lambrechts et al dataset also includes epithelial cells, we would encourage the authors to include this group as well.
- 15- In Figure 2C, the authors show ENAH expression in the single-cell RNA-sequencing dataset for human NSCLC from Lambrechts et al 2018. The authors could perform the same analysis for the published datasets for human PDAC (Bernard et al 2019, Elyada et al 2019, Peng et al 2019 Cell Research) considering that PDAC is also the focus of this work.

Referee #2:

In the present study, the authors have studied the role of hMENA in the paracrine crosstalk

between cancer cells and CAFs. The hypothesis is interesting, but the manuscript is lacking convincing evidence supporting this hypothesis.

Main Concerns

The main concern throughout the paper is the lack of biological replicates for each experiment. Some assays use a specific line of isolated CAFs which then changes to the next experiment. Including at least two biological replicates is warranted to prove that the observations are valid. Furthermore, a graphical abstract would be helpful in the different figures to show what authors are trying to prove.

Specific comments:

Fig.1 . Authors mention that FAP/aSMA are two of the main CAF markers, but FAP stainings are not shown.

Fig 1B Including more patients could further verify the observation.

Fig 1 D,E Authors show a heterogenous expression of MENAdeltaV6, but there is no quantification of expression levels. A mean level versus normal tissue or NF would be helpful.

Fig.S3B Why use the moderate expressing p#44 for the contraction assay? Why not a high delta V6 expressing CAF line?

Fig S4A From which patient were the CAFs isolated from?

Fig S4C Is the overexpression of deltaV6 of a biological level? Comparison with patient derived CAF expression would be helpful.

Fig 3A L#187 CAFs was not included in Fig.1. what was the original expression levels?

Fig 3B Only a single biological replicate is included, more replicates would help verify the observation.

Fig 3C L#310 is not in Fig.1A, P#110 has a very low expression in Fig.1A, why use it here at all?

Fig 3D,E Are the expression levels of the protein of biological relevance?

Fig 4 B Only a single biological replicate. Include more replicates of High and Low expressing CAFs.

Fig 4C Is there a significant difference of between siRNA and Culture media conditions?

Fig S5 No commas on P values, Patient CAFs different from figure 4.

Fig 4 D More replicates are needed. Do weak expressing and High expressing CAFs respond the same to PANC-1 CM or is it only P-NFs?

Fig 5A Here authors show to Normal fibroblast in replicates. Why are these fibroblasts not use in earlier assays?

Fig 5H Is there significant difference between the red and brown conditions?

Fig 6D Quantification of all proteins would be helpful to illustrate this point.

Fig 7A How does ENOH affect survival by itself? What was the cutoff for gene expression for high versus low groups?

Referee #3:

Melchionna et al., describe a link between a hMENA isoform and paracrine GAS6/AXL tumour-stroma signalling in PDAC and NSCLC. The paper is comprehensive and well written. The experiments appear robust, with appropriate controls and rescue experiments. The figures are generally clear. I congratulate the authors on an excellent piece of work.

Major Point:

Given the substantial number of cell-type specific signalling observations, the paper would greatly

benefit from a 'summary figure' illustrating the tumour-stroma reciprocal signalling axes that have been discovered. It is currently quite difficult to understand what is happening in what cell-type throughout the manuscript. A summary figure would help readers leave with a clear sense of which hMENA isoform is important in which cell - and what the downstream signalling/phenotype consequences are.

Minor Points:

Figure 2A/B - Label IHC images with the antigen being probed.

Figure 2C - Change 'EC' to 'Epithelial'.

Figure 5A - This heatmap is messy and low-resolution compared to the other figures. Can the authors redraw this to a higher quality? Also, the legend requires a scale (is it z-score?). The colours cannot be observed by those with red-green colour blindness - please re-plot using blue/red (or equivalent).

Figure 6D - The labels at the bottom of this blot (describing the Gas6 treatments) should be moved to the top so the reader can easily observe the experimental variables before viewing the results.

Page 10 - 'secretoma' should be 'secretome'?

LC-MS/MS data should be made available on PRIDE (<https://www.ebi.ac.uk/pride/>).

** As a service to authors, EMBO Press provides authors with the ability to transfer a manuscript that one journal cannot offer to publish to another journal, without the author having to upload the manuscript data again. To transfer your manuscript to another EMBO Press journal using this service, please click on
Link Not Available

Dear Dr Senyilmaz-Tiebe
Editor of EMBO Reports

According to your suggestions and our correspondence we are submitting a revised version of the manuscript "The actin modulator hMENA regulates GAS6-AXL axis and pro-tumor cancer/stromal cells cooperation" to be reconsidered for publication in EMBO Reports.

We thank the Reviewers for the helpful comments and suggestions, all of which have greatly improved and strengthened our manuscript. We have duly performed all the experiments necessary to respond to the Reviewers' criticisms and have modified the manuscript accordingly (changes are marked up).

Below, is a point-by-point response to the Reviewers' comments.

We hope that our revised paper meets with your final approval and look forward to hearing from you soon.

Yours sincerely,
Paola Nisticò, M.D.

Point-by-point response

Referee #1

1) The authors should better characterize the CAF lines generated to make sure that they are not EMT cells. Especially considering the previous description of pro-invasive hMENA Δ v6 in E-cadherin-low mesenchymal cells (di Modugno 2012). For example, in addition to alpha-SMA, which can be expressed by EMT cells, the authors should check other fibroblast markers by IF (e.g. FAP, PDGFR). Additionally, the authors should confirm that the CAF lines used do not carry any mutation present in the tumour sample of origin (e.g. KRAS, EGFR). A characterization of bona fide CAFs would strengthen the significance of this study.

We thank the Reviewer for these comments and accordingly we extensively characterized a panel of CAF cultures (n=11) including new CAF preparations. All functional studies have now been performed in these CAFs, after an in depth characterization.

Firstly, to rule out the possibility that the isolated CAF cultures did not contain EMT cancer cells we performed multiple gene mutation analysis with a 22-gene panel (EGFR, ALK, ERBB2, ERBB4, FGFR1, FGFR2, FGFR3, MET, DDR2, KRAS, PIK3CA, BRAF, AKT1, PTEN, NRAS, MAP2K1, STK11, NOTCH1, CTNNA1, SMAD4, FBXW7,

TP5) by using NGS technology. As reported in Appendix Table 2 NSCLC tumors, which were mutated in genes such as EGFR, p53 and KRAS, did not show any mutations in the paired isolated CAFs. Similar results (Appendix Table 1) have been obtained in P-CAF. (First paragraph of the results of revised version).

By using RNA obtained from the same samples analyzed by NGS we performed qRT-PCR showing the expression of PDGFR stromal marker, but not of the epithelial cell adhesion molecule (EpCAM) (New Figure S1B,C). Finally, as also suggested by Reviewer 2, we have evaluated FAP expression by confocal immunofluorescence (new Figure S1A).

2-Although we appreciate that the authors show for all experiments at least 2 technical repeats, the majority of experiments are only shown for 1 PDAC CAF line, and for 1 NSCLC CAF line. The authors should show at least 2 biological replicates (i.e. 2 different CAF lines) for all experiments shown.

We agree with the Reviewer and the lack of biological replicates in our previous version was mainly due to the limitations in non-immortalized CAF primary culture manipulation. In the new version we were able to perform all the functional experiments in 2 selected PDAC (P-CAF#36 and #138) and 2 NSCLC CAFs (L-CAF#189 and #484) with high hMENA Δ v6 expression and 2 PDAC (P-CAF#44 and 110) and 1 LUNG CAF (L-CAF#400) with low hMENA/hMENA Δ v6 expression. The loss and gain of function experiments reported in the new Figures 3 and S6 have been done in these CAFs and in normal lung fibroblasts purchased from ATCC.

3- Similar to the point above, for all relevant experiments, the authors should use a second PDAC cancer line in addition to Panc-1, and a second NSCLC line in addition to A549 (the authors do have H1975 and H1650 cells, but only use them once).

In agreement with the Reviewer's suggestion we have done relevant experiments shown in the new Figures 4C, D, S7A and C by using additional PDAC (KP-4) and NSCLC (H1975) cell lines.

4- We appreciate that the authors have used several CAF lines throughout the paper. However, a major concern is the fact that no single CAF line has been used for all different experiments shown (e.g. in Figure 3, three different lung CAF lines have been used for three different experiments). This lack of consistency is a significant weakness of the paper, as it is difficult to assess whether the conclusions made by the authors are generalizable to various PDAC and NSCLC CAFs or are due to cell line-specific features. To support their claims, the authors should perform all experiments with at least the same 2 PDAC CAF lines and the same 2 lung CAF lines. Inclusion of these data would significantly strengthen the authors' conclusions.

We again thank the Reviewer who suggested a more rigorous approach. Accordingly, as above mentioned, all functional studies reported in the paper have now been conducted in the same P-CAF and L-CAF. All the knockdown experiments were performed with 2 PDAC CAFs (#36 and 138) and 2 LUNG CAFs (#186 and 484) with high hMENA/hMENA Δ 6 expression. In parallel, we performed all gain of function

experiments using 2 PDAC CAFs (#44 and 110) and NSCLC CAFs# 400 with low levels of hMENA/hMENA Δ v6. All these experiments are shown in the new Figures 3 and S6.

5-In Figure 5, the authors use P#110 CAFs and L#182 CAFs for hMENA knock-down experiments. However, in Figures 1D and 1E, the levels of hMENADv6 are very low in these CAF lines. The authors should justify why they chose these CAF lines and explain why they think they are more relevant than using CAF lines with higher hMENA basal levels.

We apologize and in agreement with the Reviewer's suggestion we have silenced hMENA/ hMENA Δ v6 in P#138 and L-CAF#189 (highly expressing) and analyzed the GAS6 expression. The new results are shown in the revised Figures 5F and G.

6- The authors show that a high hMENA/GAS6/AXL signature is associated with poor prognosis in PDAC and NSCLC. However, in their previous publications, as also pointed out in the introduction, they have already shown that hMENA alone is associated with poor prognosis in cancer. The authors should compare the 2 signatures and comment on why it is relevant/novel to point out that a high HENA/GAS6/AXL signature is associated with poor prognosis.

We have clarified this complex issue and added in Figures 7 and S12 the survival curves relative to ENAH expression alone showing that ENAH alone is not prognostic, while the 3-gene (ENAH, AXL and GAS6) expression signature is associated with poor prognosis in PDAC and NSCLC patients. Our previous studies (Bria et al, 2014, Melchionna et al, 2016, Di Modugno et al, 2018b), cited in the introduction, indicated a prognostic algorithm based on differential hMENA isoform expression by using a combination of IHC staining with two antibodies (pan-hMENA which recognizes all hMENA isoforms and the specific hMENA11a mAb). The use of the single pan-hMENA Ab was not predictive of survival also in our previous studies, indicating that only the pattern of isoform expression, not detectable by the gene expression analysis, may be predictive of survival.

7- In Figure 2, the authors identify potential CAFs in tissues due to their elongated morphology. However, this is not sufficient to identify CAFs. To corroborate their point, the authors should perform co-IF with hMENA and FAP or, at least, alpha-SMA. Additionally, the authors should do this co-IF for PDAC tissues, as most of the in vitro data presented focus on PDAC CAF lines.

Thank you for this suggestion and now we have added a new Figure 2 (A and B) showing that hMENA is expressed in alpha-SMA positive cells both in primary NSCLC and PDAC tissues, as defined by confocal analysis of paraffin-embedded tissues. The primary tumors analyzed were relative to hMENA Δ v6 high CAF cultures.

8- The authors should include the knock-down and over-expression validation (qPCR and/or western blot) for all cell lines used, and clarify which lines they show in Figures S4A and S4B. Also, the authors use some CAF lines whose hMENADv6 levels are not shown in Figures 1D and 1E. The authors should include these data as well.

To clearly show the efficiency of knockdown and the overexpression of hMENA/hMENA Δ v6 in all CAFs used in functional experiments, the relative western

blot are shown in the new Figure 3 and we have removed the previous Figures S4A and B.

9- The authors should include additional relevant references. For example, at the beginning of the introduction, when mentioning the role of the stroma in "hampering treatment efficacy", the authors only cite Provenzano et al, but several other papers should be cited (see David Tuveson, Rakesh Jain, Ronald Evans, etc work). Additionally, at the beginning of the discussion the authors state that "many studies" have pointed out the cancer/CAF signalling but only cite Tape et al. Indeed, many recent studies on PDAC have contributed to this discussion and should be acknowledged (see Tony Hunter, David Tuveson, Ben Stanger, etc work).

As suggested by the Reviewer we have included the requested references (highlighted in yellow) in the introduction (Kraman et al Science 2010; Shi et al Nature 2019) and discussion (Sahai et al Nat. Rev. Cancer 2020) sections.

Minor Points

- 1- Please consider the journal's guidelines before finalizing the manuscript (e.g. the title should not include not commonly used abbreviations and it should not be longer than 100 characters; the table legends are missing and should be included).**

We have shortened the title and included the table legends.

- 2- Please use correct nomenclature for gene names (i.e. human genes/mRNA should be in capital and italics).**

We have corrected the nomenclature.

- 3- Please define all acronyms the first time they are introduced (e.g. TGF-beta, EGFR).**

We have defined the acronyms.

- 4- Please make sure that all scale bars are big enough to be clearly visible.**

We have modified the scale bars.

- 5- Please include pan-hMENA blot for Figures 1D and 1E. This is relevant since in some knock-down assays the authors use CAF lines with really low levels of hMENADv6, and it is unclear why using these lines is relevant (see major point #5 above).**

Data relative to the previous Figure 1D and 1E are now shown in Figure 1B and C which include Pan-hMENA blot and quantification, as reported in major point #5.

6- The authors should clarify whether the Student's t test used is unpaired or paired.

We have clarified this point in the method section relative to statistical analysis and indicated that we used an unpaired Student's t test.

7- What is the effect of hMENA knock-down and overexpression on the growth and alpha-SMA levels of CAFs? This is relevant as the authors claim that hMENA has a crucial role in CAF activation. Similarly, do P-CAF High proliferate at the same rate of P-CAF Low?

We did not observe a reduction of α -SMA expression following hMENA silencing in our CAFs, probably due to our 2D experimental conditions. We would like to assess this point in the future by using 3D culture conditions. We have performed preliminary experiments to understand the role of hMENA in CAF proliferation and we did not find a significant difference in P-CAF hMENA silenced, at least in our basal culture conditions without pro-proliferative stimuli. We believe that to be conclusive the results of hMENA effect on CAF proliferation, deserve a deeper investigation.

8- In Figure 4A, the authors should indicate which CAF line corresponds to which column, as they only use a few lines for their following experiments. The authors should also clarify whether P-NFs corresponds to P-NF1 and P-NF2 or other lines.

We have indicated which CAF line corresponds to which column, in the new Figure 1B that now includes the new P-CAF#138.

P-NF corresponds to P-NF1 which was used for all functional studies with the exception of LC-MS/MS analysis which included also the P-NF2.

9- The authors should at least justify why in different CM experiments they use 24h or 48h timepoints (e.g. for Panc1 growth and invasion assays). Similarly, they should clarify why they use transwells in some cases and CM in others. Consistent experimental conditions would allow to compare different assays more appropriately.

The different timepoints of CM stimulation were used considering the different endpoints (i.e. invasion and growth). To clarify this issue: 1) we assessed the effect of CAF-secretome on tumor cell invasiveness, by treating the tumor cells for 48 hrs with CAF-CM derived from CNT and/or hMENA silenced CAFs. Then the matrigel invasion assay was conducted for 16 hrs. 2) We assessed the effect of CAF CM on tumor cell growth, by treating tumor cells with CAF-CM derived from CNT and/or hMENA silenced CAFs for 24h. In the Figure S8 we showed that the addition of CAF-derived CM for 24hrs is able to increase PANC1 viability.

The indirect co-culture system (with a transwell pore of 0,4 um) was used when we looked at the role of hMENA in the cooperation between cancer cells and CAFs.

10- We suspect that Figure S5 (right panel) may have been mistakenly labelled (x axis), and that it refers to Panc-1 cells.

We apologize for the mistake and x axis is now correctly labeled.

11- In Figures S6A and S6B, the authors should clarify which P-CAF line is shown.

We have included the P-CAF patient number as requested by the Referee in the new Figure 1B.

Not addressing the following points will not preclude this manuscript from publication, but we find they would be useful for the readership.

12- In Figures 1B and 1C, please indicate which hMENA isoform corresponds to which band as done for Supp Fig S6.

We have indicated to which bands the hMENA isoforms correspond.

13- The authors show IF staining for pan-hMENA and hMENA11a. As showing the mesenchymal-specific isoform would be relevant, the authors should also show, if possible, IF and IHC with the hMENADv6 antibody (for Figure 1 and 2). We, however, appreciate that this may not be possible for technical reasons. If so, we would encourage the authors to mention this somewhere in the manuscript.

Unfortunately, as reported in our previous work describing the hMENA Δ v6 isoform (Di Modugno et al PNAS 2012) the hMENA Δ v6 antibody is not suitable for either IHC or IF analysis.

14- In Figure 2C, the authors show increased ENAH expression in fibroblasts compared to other stromal cells. Since the Lambrechts et al dataset also includes epithelial cells, we would encourage the authors to include this group as well.

In agreement with the Reviewer's suggestion we have included the epithelial and alveolar groups in the new Figure 2D.

15- In Figure 2C, the authors show ENAH expression in the single-cell RNA-sequencing dataset for human NSCLC from Lambrechts et al 2018. The authors could perform the same analysis for the published datasets for

human PDAC (Bernard et al 2019, Elyada et al 2019, Peng et al 2019 Cell Research) considering that PDAC is also the focus of this work.

We were able to analyze the Peng cohort and the relative data (shown in the new Figure 2C), confirming the ENAH expression in CAFs from human PDAC, are reported in methods, results and discussion sections.

Referee #2

Main Concerns

The main concern throughout the paper is the lack of biological replicates for each experiment. Some assays use a specific line of isolated CAFs which then changes to the next experiment. Including at least two biological replicates is warranted to prove that the observations are valid.

Furthermore, a graphical abstract would be helpful in the different figures to show what authors are trying to prove.

We agree with the Reviewer who raised similar concerns to the Reviewer 1. The lack of biological replicates in our previous version was mainly due to the limitations in non-immortalized CAF primary culture manipulation. Their comments helped us to perform more rigorous methodological approaches. In the new version we were able to perform all the functional experiments in 2 selected PDAC (P-CAF#36 and #138) and 2 NSCLC CAFs (L-CAF#189 and #484) with high hMENA Δ v6 expression and 2 PDAC (P-CAF#44 and 110) and 1 LUNG CAF (L-CAF#400) with low hMENA/hMENA Δ v6 expression. All the loss and gain experiments reported in the new Figures 3 and S6 have been done in these CAFs and in lung normal fibroblasts purchased from ATCC.

In this revised version we have included, as suggested also by Reviewer 3 a new Figure 8, which illustrates the working model of the key findings.

Specific comments:

Fig.1 Authors mention that FAP/aSMA are two of the main CAF markers, but FAP staining are not shown.

Thank you for this suggestion. In the new Figure S1A we show representative images of the confocal analysis of FAP expression of all the CAFs used for functional studies.

Fig 1B Including more patients could further verify the observation.

In the new Figure 1 we have shown CAF preparation relative to more patients and the same CAFs were used for functional studies. Quantification of hMENA Δ v6 expression has been reported in this new figure.

Fig 1 D, E Authors show a heterogenous expression of MENAdeltaV6, but there is no quantification of expression levels. A mean level versus normal tissue or NF would be helpful.

The quantification of hMENAΔv6 expression in L-CAFs is now reported in Figure 1C where we compared the hMENAΔv6 expression with respect to normal lung fibroblasts purchased by ATCC.

Fig.S3B Why use the moderate expressing p#44 for the contraction assay? Why not a high delta V6 expressing CAF line?

We apologize and in agreement with the suggestions we have done all the functional experiments, including the contraction assays in Normal fibroblasts and high hMENAΔv6 expressing CAFs (P-CAF#138 and L-CAF#189) as shown in new Figure S5B.

Fig S4A From which patient were the CAFs isolated from?

The previous Figures S4A and B were removed and data relative to the efficiency of knockdown and overexpression of hMENA/hMENAΔv6 in all CAFs used in functional experiments and the relative western blot are shown in the new Figure 3 where the CAF number refers to the relative patient.

Fig S4C Is the overexpression of deltaV6 of a biological level? Comparison with patient derived CAF expression would be helpful

Again the data relative to this figure has been replaced by testing a new panel of CAFs and the biological effect of hMENAΔv6 overexpression has now been compared among normal fibroblasts, low hMENAΔv6 CAF and high hMENA by gain and loss of function experiments (Figures 3 and S6).

Fig 3A L#187 CAFs was not included in Fig.1. what was the original expression levels?

We have now eliminated the L-CAF#187 and the new Figure 1 shows all the new CAF cultures as explained in response to your main concerns.

Fig 3B Only a single biological replicate is included, more replicates would help verify the observation.

Fig 3D,E Are the expression levels of the protein of biological relevance?

Fig 4 B Only a single biological replicate. Include more replicates of High and Low expressing CAFs.

We agree with these observations in part raised also by the Reviewer 1, and in the new version we show more replicates and the data clearly sustain the role of hMENAΔv6 expression in CAF activation (new Figures 3 and S6).

Fig 4C Is there a significant difference of between siRNA and Culture media conditions?

We found a significant difference only between culture medium conditions and siCNT as well as between siCNT- and sihMENA(t)-CAF-CM, but not between culture medium and sihMENA-CAF-CM. The significant statistical differences are indicated in the new Figure 4B.

Fig S5 No commas on P values, Patient CAFs different from figure 4.

We apologize for these mistakes and we have duly corrected them.

Fig 4 D More replicates are needed. Do weak expressing and High expressing CAFs respond the same to PANC-1 CM or is it only P-NFs?

As suggested we have done more replicates and data are reported in the new Figure S7B as well as in the results section. Additionally, we performed similar experiments using high hMENA Δ v6 expressing CAFs derived from NSCLC stimulated with CM of the H1975 NSCLC cell line (Figure S7B).

Fig 5A Here authors show to Normal fibroblast in replicates. Why are these fibroblasts not use in earlier assays?

We were limited in the use of the two P-NF preparations due to the difficulties to expand this culture.

Fig 5H Is there significant difference between the red and brown conditions

P value of 0.03 between these two groups was found by using a non-adjusted t test. We do not show the significance in the Figure as the P value was not significant when we used the one-way ANOVA, followed by Bonferroni's multiple comparison test.

Fig 6D Quantification of all proteins would be helpful to illustrate this point.

The quantification of AKT phosphorylation level upon GAS6 treatment in control and hMENA silenced PANC-1 has been included in this figure.

Fig 7A. How does ENOH affect survival by itself? What was the cutoff for gene expression for high versus low groups?

We include the survival curves for hMENA that clearly indicated that hMENA signature by itself does not affect patient survival (New Figures 7A-B and Figure S12).

Patients were stratified into two groups on the basis of the signature expression levels using quartiles as thresholds, and the best fits are reported. In detail, for

PDAC patients were reported the curves using as cut-off the 75th percentile of the signature expression and for LUSC patients were reported the curves using as cut-off the 25th percentile of the signature expression.

Referee #3

Melchionna et al., describe a link between a hMENA isoform and paracrine GAS6/AXL tumour-stroma signalling in PDAC and NSCLC. The paper is comprehensive and well written. The experiments appear robust, with appropriate controls and rescue experiments. The figures are generally clear. I congratulate the authors on an excellent piece of work.

Major Point:

Given the substantial number of cell-type specific signalling observations, the paper would greatly benefit from a 'summary figure' illustrating the tumour-stroma reciprocal signalling axes that have been discovered. It is currently quite difficult to understand what is happening in what cell-type throughout the manuscript. A summary figure would help readers leave with a clear sense of which hMENA isoform is important in which cell- and what the downstream signalling/phenotype consequences are.

First of all, we thank the Reviewer for the congratulations which we found very encouraging.

As suggested we have included in the new version a summary figure with the key findings of the manuscript (New Figure 8).

Minor Points:

Figure 2A/B - Label IHC images with the antigen being probed.

We have labeled the IHC images with the appropriate antigen probed in the new Figure S3.

Figure 2C - Change 'EC' to 'Epithelial'.

EC in the figure refer to endothelial cells. We also introduced in this panel the epithelial cells (Ep) in panel D of the new Figure 2, where we now also show data relative to the Peng PDAC dataset (panel C).

Figure 5A - This heatmap is messy and low-resolution compared to the other figures. Can the authors redraw this to a higher quality? Also, the legend

requires a scale (is it z-score?). The colours cannot be observed by those with red-green colour blindness - please re-plot using blue/red (or equivalent).

Thanks to these suggestions, we have modified the colors and reported in the legend the scale, which is a log2 scale.

Figure 6D - The labels at the bottom of this blot (describing the Gas6 treatments) should be moved to the top so the reader can easily observe the experimental variables before viewing the results.

We moved the labels to the bottom of this blot.

Page 10 - 'secretoma' should be 'secretome'?

Corrected. Thanks.

LC-MS/MS data should be made available on PRIDE (<https://www.ebi.ac.uk/pride/>)

*The data linked to MassIVE repository are available following this link: <https://massive.ucsd.edu/ProteoSAFe/static/massive.jsp> and using the following access ID: **MSV000084685**. We are unable to link the raw data to PRIDE, since our co-author Emily Chen has moved from Columbia University.*

Dear Paola,

Thank you for submitting the revised version of your manuscript. It has now been seen by all of the original referees.

As you can see, the referees find that the study is significantly improved during revision and recommend publication. Please note that only two of the referees submitted a report, but the third referee just let us know that his/her concerns were satisfactorily addressed and she/he recommends publication. Before I can accept the manuscript, I need you to address some minor points below:

- We noted that the resolution/quality of some figures are not high enough - e.g. Figure 1A, Figure 6A, Figure S1A. Please provide higher resolution figures.
- Please fill out and include an author checklist as listed in our online guidelines (<https://www.embopress.org/page/journal/14693178/authorguide>)
- As of January 2016, new EMBO Press policy asks for corresponding authors to link to their ORCID iDs. You can read about the change under "Authorship Guidelines" in the Guide to Authors here: <http://emboj.embopress.org/authorguide>

In order to link your ORCID iD to your account in our manuscript tracking system, please do the following:

1. Click the 'Modify Profile' link at the bottom of your homepage in our system.
2. On the next page you will see a box halfway down the page titled ORCID*. Below this box is red text reading 'To Register/Link to ORCID, click here'. Please follow that link: you will be taken to ORCID where you can log in to your account (or create an account if you don't have one)
3. You will then be asked to authorise Wiley to access your ORCID information. Once you have approved the linking, you will be brought back to our manuscript system.

We regret that we cannot do this linking on your behalf for security reasons.

- We noted that the Appendix Table callouts are missing the 'S' i.e. Appendix Table S1 and S2.
- The Table of Contents of the Appendix file is currently missing page numbers. Each figure needs to be called 'Appendix Figure S#'
- As per format requirements 'graphical abstracts' are not allowed. Please remove the callout to Figure 8 from the text. Please consider converting Figure 8 into a 'synopsis image'. We note that the image is currently quite detailed and the labels of the image cannot be read well when resized to 550x400 pixels (which will be its final size when published online). Please edit and simplify the image accordingly.
- Papers published in EMBO Reports include a 'Synopsis' to further enhance discoverability. Synopses are displayed on the html version of the paper and are freely accessible to all readers. The synopsis includes a short standfirst summarizing the study in 1 or 2 sentences that summarize the key findings of the paper and are provided by the authors and streamlined by the handling editor. I would therefore ask you to include your synopsis blurb.
- Our production/data editors have asked you to clarify several points in the figure legends (see attached document). Please incorporate these changes in the attached word document and return it with track changes activated.

Thank you again for giving us to consider your manuscript for EMBO Reports, I look forward to your minor revision.

Kind regards,

Deniz

--

Deniz Senyilmaz Tiebe, PhD
Editor
EMBO Reports

Referee #1:

We appreciate the authors' efforts in addressing our concerns, in particular considering the timeframe provided and these challenging times. We consider the revised manuscript acceptable for publication in EMBO Reports.

As a minor point, we think that the authors should clarify in the figure legends which CAF lines have been used for the experiments shown (e.g. in Fig 5H and S8 this information appears missing).

Referee #2:

The authors have really made an effort to respond to our previous comments. As I can see, all of our concerns in the first round of review has been addressed properly and the manuscript can now be considered for publication.

Dear Dr Deniz Senyilmaz-Tiebe
Editor of EMBO Reports

Rome, August 3rd 2020

According to your comments we are submitting a revised version of the manuscript "The actin modulator hMENA regulates GAS6-AXL axis and pro-tumor cancer/stromal cell cooperation".

The resolution quality of the figures has been ameliorated.

An Author checklist and my ORCID have been included.

Highlighted in the figure legends the modification requested also by the Reviewer 1.

Figure 8 has been removed and replaced by the synopsis.

We hope that our revised paper meets with your final approval and look forward to hearing from you soon.

Yours sincerely,
Paola Nisticò, M.D.

Dear Paola,

Thank you for submitting your revised manuscript. I have now looked at everything and all is fine. Therefore I am very pleased to accept your manuscript for publication in EMBO Reports.

Congratulations on a nice work!

I have noted that the labels of the synopsis image are hard to read when resized to 550 pixels wide, which will be its final size when published online (please see attached). Could you please make the labels in the marked box bigger/thicker? You can send the file per email.

Kind regards,

Deniz

--

At the end of this email I include important information about how to proceed. Please ensure that you take the time to read the information and complete and return the necessary forms to allow us to publish your manuscript as quickly as possible.

As part of the EMBO publication's Transparent Editorial Process, EMBO reports publishes online a Review Process File to accompany accepted manuscripts. As you are aware, this File will be published in conjunction with your paper and will include the referee reports, your point-by-point response and all pertinent correspondence relating to the manuscript.

If you do NOT want this File to be published, please inform the editorial office within 2 days, if you have not done so already, otherwise the File will be published by default [contact: emboreports@embo.org]. If you do opt out, the Review Process File link will point to the following statement: "No Review Process File is available with this article, as the authors have chosen not to make the review process public in this case."

Should you be planning a Press Release on your article, please get in contact with emboreports@wiley.com as early as possible, in order to coordinate publication and release dates.

Thank you again for your contribution to EMBO reports and congratulations on a successful publication. Please consider us again in the future for your most exciting work.

THINGS TO DO NOW:

You will receive proofs by e-mail approximately 2-3 weeks after all relevant files have been sent to our Production Office; you should return your corrections within 2 days of receiving the proofs.

Please inform us if there is likely to be any difficulty in reaching you at the above address at that time. Failure to meet our deadlines may result in a delay of publication, or publication without your

corrections.

All further communications concerning your paper should quote reference number EMBOR-2020-50078V3 and be addressed to emboreports@wiley.com.

Should you be planning a Press Release on your article, please get in contact with emboreports@wiley.com as early as possible, in order to coordinate publication and release dates.

Corresponding Author Name: Paola Nisticò

Journal Submitted to: Embo reports

Manuscript Number: EMBOR-2020-50078V2-Q